# Gametes deficient for Pot1 telomere binding proteins alter levels of telomeric foci for multiple generations

Evan H. Lister-Shimauchi[1,2,5✉], Michael Dinh[1,2,5], Paul Maddox [2,3,4] & Shawn Ahmed [1,2,3,4✉]

Deficiency for telomerase results in transgenerational shortening of telomeres. However, telomeres have no known role in transgenerational epigenetic inheritance. *C. elegans* Protection Of Telomeres 1 (Pot1) proteins form foci at the telomeres of germ cells that disappear at fertilization and gradually accumulate during development. We find that gametes from mutants deficient for Pot1 proteins alter levels of telomeric foci for multiple generations. Gametes from *pot-2* mutants give rise to progeny with abundant POT-1::mCherry and mNeonGreen::POT-2 foci throughout development, which persists for six generations. In contrast, gametes from *pot-1* mutants or *pot-1; pot-2* double mutants induce diminished Pot1 foci for several generations. Deficiency for MET-2, SET-25, or SET-32 methyltransferases, which promote heterochromatin formation, results in gametes that induce diminished Pot1 foci for several generations. We propose that *C. elegans* POT-1 may interact with H3K9 methyltransferases during *pot-2* mutant gametogenesis to induce a persistent form of transgenerational epigenetic inheritance that causes constitutively high levels of heterochromatic Pot1 foci.

[1] Department of Genetics, University of North Carolina, Chapel Hill, NC 27599, USA. [2] Department of Biology, University of North Carolina, Chapel Hill, NC 27599, USA. [3] Lineberger Comprehensive Cancer Center, University of North Carolina, Chapel Hill, NC 27599, USA. [4] Curriculum in Genetics and Molecular Biology, University of North Carolina, Chapel Hill, NC 27599, USA. [5] These authors contributed equally: Evan H. Lister-Shimauchi, Michael Dinh.
✉email: evanhlister@gmail.com; shawn@med.unc.edu

Most eukaryotic chromosome ends are composed of a tract of double-stranded tandem DNA repeats that terminates with a 3′ single-stranded overhang. Telomeric proteins prevent chromosome termini from being recognized as an aberrant form of DNA damage and protect the ends of linear chromosomes from nucleolytic attack. Telomeres normally erode as a consequence of incomplete DNA replication. However, telomere length is maintained by telomerase, which utilizes TElomerase Reverse Transcriptase (TERT) and telomerase RNA core subunits to add telomere repeats to chromosome termini. Telomere length is maintained across generations by high expression levels of TERT in germ cells[1]. Deficiency for telomerase in human families results in transgenerational shortening of telomeres, which causes an earlier age of onset of disease in successive generations[2,3].

Ciliated protozoans that possess abundant telomeres enabled biochemical identification of single-stranded telomere binding proteins[4–8], which were later shown to be functionally homologous to Pot1 proteins from *Schizosaccharomyces pombe* and humans[9]. Knockout of *S. pombe* Pot1 results in rapid telomere dysfunction[9]. Mammalian POT1 is part of a six subunit protein complex, termed shelterin, which interacts with telomeres to promote telomere stability and prevent chromosome termini from being recognized as DNA damage[10,11].

Shelterin protects telomeres from being fused together by facilitating the formation of a T-loop at chromosome termini, where a strand invasion intermediate forms when the single-stranded 3′ telomeric overhang intercalates into a segment of double-stranded telomeric DNA[12]. Aside from 3′ single-stranded telomeric overhangs that are created in the context of telomere replication, POT1 may interact with single-stranded telomeric DNA that is displaced by the T-loop. In addition, a subset of nuclear POT1 may associate with shelterin complexes that interact with double-stranded telomeric DNA via the TRF1 and TRF2 Myb-domain proteins. That said, POT1 is 10-fold less abundant than some shelterin subunits and may therefore be enriched at segments of single-stranded telomeric DNA[13].

Expression of POT1 in human cells results in telomere elongation[14,15]. Consistently, the shelterin subunits POT1 and TPP1 have been demonstrated to promote telomerase activity in vitro[14,16,17]. However, POT1 can also negatively regulate the activity of telomerase in vitro[18], suggesting that POT1 may possess opposing telomeric functions.

The human genome contains a single POT1 gene that encodes a protein with two OB-fold domains, OB1 and OB2, which interact with single-stranded telomeric DNA. Mice, *Arabidopsis*, and *Tetrahymena* have multiple Pot1 genes that possess one or two OB-fold domains[19–21]. The nematode *Caenorhabditis elegans* possesses four Pot1 genes that encode single OB-fold domains: *pot-1*, *pot-2*, *pot-3*, and *mrt-1*[22]. *C. elegans* POT-1 has an OB1 fold, whereas POT-2, POT-3, and MRT-1 each encode an OB2 fold[23]. Both POT-1 and POT-2 proteins can promote T-loop formation in vitro[23]. Mutation of either *pot-1* or *pot-2* leads to gradual lengthening of telomeres over ~16 generations, which depends on telomerase[22]. This phenotype is not exacerbated by simultaneous mutation of both *pot-1* and *pot-2*[22], implying that these proteins may function together to repress telomerase. However, POT-1 and POT-2 proteins have distinct in vitro affinities for 5′ and 3′ ssDNA overhangs, respectively[23]. Furthermore, POT-1 but not POT-2 promotes tethering of telomeres to the nuclear periphery[24], and *C. elegans trt-1* telomerase mutants that are deficient for *pot-2* but not *pot-1* display an increased incidence of the telomerase-independent telomere maintenance pathway termed Alternative Lengthening of Telomeres (ALT)[25,26]. Together, these observations suggest that *C. elegans* POT-1 and POT-2 have distinct and common functions.

A third *C. elegans* protein MRT-1 contains an OB2 fold that is required for telomerase activity in vivo[27], consistent with the proposed role for mammalian POT-1 in telomerase function[16,17].

Aside from shelterin, mammalian telomeric DNA interacts with histones that are enriched for H3K9 di- and trimethylation that promote genome silencing[28]. *C. elegans* telomeres also possess the H3K9 dimethyl histone silencing mark[29], suggesting that heterochromatin may be a common theme of metazoan telomeres. Loss of heterochromatin in mouse primary cells deficient for the histone methyltransferases Suv39h1 and Suv39h2 is associated with telomere elongation[28].

In this report, we uncover a novel connection between epigenetic inheritance and nuclear foci formed by telomere binding proteins. We find that gametes from mutants that lack *C. elegans* POT-1 or POT-2 single-stranded telomere binding proteins induce altered levels of telomeric foci for multiple generations. Histone H3 methyltransferases with known roles in transgenerational epigenetic inheritance also alter levels of telomeric foci. As regulation of telomeres and Pot1 have been tied to aging and cancer[30,31], transgenerational epigenetic inheritance of Pot1 foci may be relevant to human health.

## Results

**Pot1 foci increase during embryonic development.** We previously created a single-copy transgene that expresses POT-1::mCherry and observed nuclear mCherry foci at telomeres of adult *C. elegans* germ cells as well as weak mCherry fluorescence throughout the nucleoplasm[22]. The specificity of POT-1::mCherry to telomeres of meiotic germ cells was supported by a predictable change in the number of POT-1::mCherry foci in response to an altered number of chromosomes[22]. We investigated the dynamics of POT-1 protein expression and observed strong POT-1::mCherry expression in mature sperm[22]. By contrast, the three most proximal (mature) oocytes displayed a reduced number of POT-1::mCherry foci (Supplementary Fig. 1a). Surprisingly, fertilization of oocytes resulted in 1-cell embryos that lacked POT-1 foci in both parental pronuclei and also in zygotic interphase nuclei (Fig. 1a). The number of POT-1::mCherry foci gradually increased during embryonic development to 8.0 ± 0.5 foci per nucleus in 32-cell embryos (Fig. 1b). We were able to quantify POT-1::mCherry foci in a small fraction of germ cells at L1–L4 larval stages and observed an abrupt transition where the number of foci doubled as L4 larvae matured into adults (Supplementary Fig. 1c, d). Therefore, POT-1 foci vanish in freshly fertilized embryos but accumulate during embryonic development and then undergo a pronounced transition at the onset of adulthood.

We previously showed that deficiency for *pot-2* did not affect the abundance of POT-1::mCherry foci in meiotic cells of the adult germline[22]. Although wild-type animals had a diminished number of POT-1::mCherry foci in the most proximal oocytes (1.4 ± 1.3), *pot-2* mutants had 11.4 ± 1.1 POT-1::mCherry foci per nucleus (Supplementary Fig. 1b, Wilcox $p < 0.001$). This corresponds to the number of chromosome ends expected for the 6 paired homologous chromosomes that are present in *C. elegans* oocytes. Abundant POT-1::mCherry foci were also observed in pronuclei and interphase nuclei of 1-cell *pot-2* mutant embryos (15.2 ± 4.7 foci per nucleus) (Fig. 1a), suggesting that POT-2 normally dismantles POT-1 foci in 1-cell embryos. Mutation of *pot-2* causes gradual telomere elongation over the course of 20 generations[22], and the *pot-2* mutant strain that we initially studied for expression of POT-1::mCherry foci had been cultured for more than 50 generations prior to analysis. We therefore utilized a strain in which the *pot-2(tm1400)* mutation had been outcrossed 14 times and possessed telomeres whose lengths were

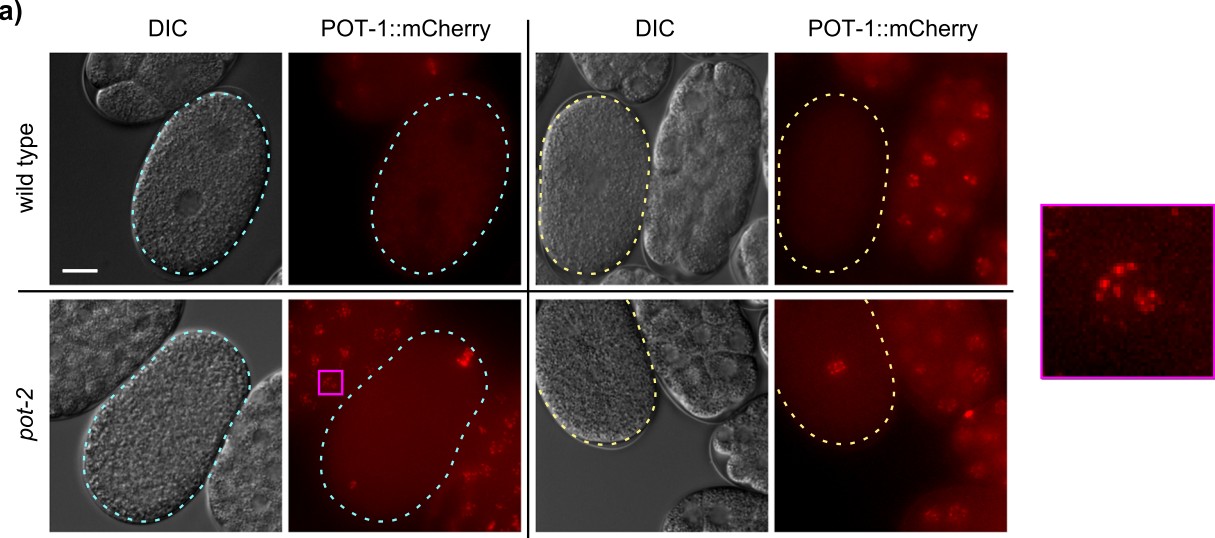

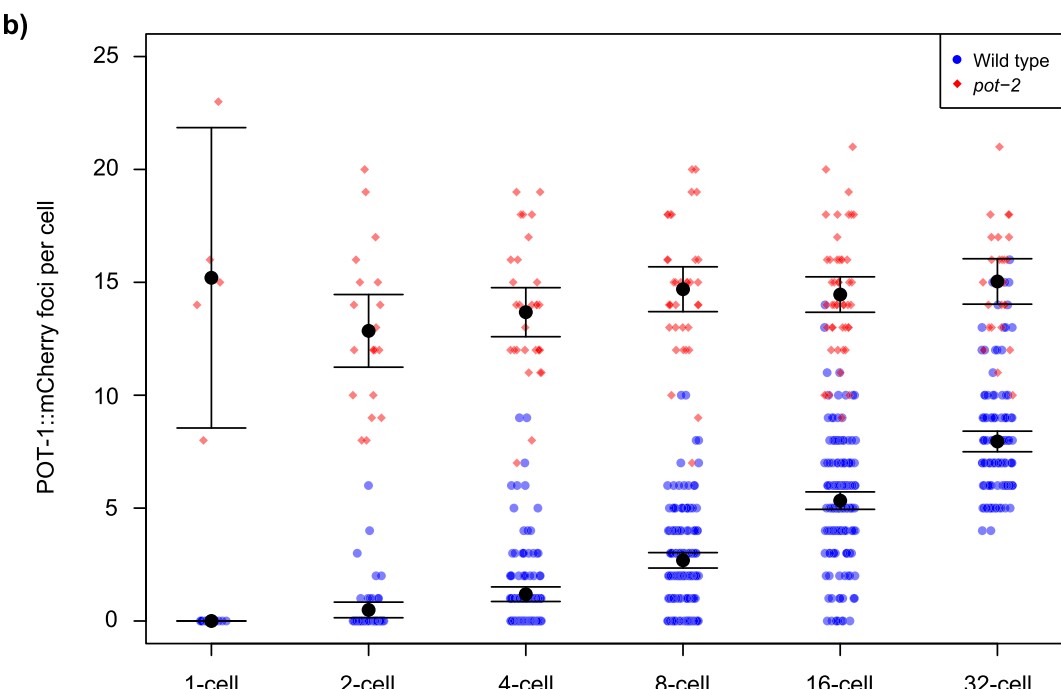

**Fig. 1 Pot1 foci increase during early embryonic development. a** DIC and fluorescent images of 1-cell embryos before and after pronuclear fusion in either wild-type or *pot-2* mutant strains. Cyan and yellow dashed lines indicate 1-cell embryos before and after pronuclear fusion, respectively. Scale bar is 10 μm. The portion of the image indicated by the magenta square is shown at 10x higher detail to the right. **b** POT-1::mCherry foci per nucleus of individual cells of embryos from 1- to 32-cell stage in wild-type (blue circles) and *pot-2* mutants (red diamonds). The difference between wild-type and *pot-2* mutants is significant at every stage (Wilcox $p < 10^{-5}$). Error bars are 95% confidence intervals. Central black dots are means.

only modestly longer than those of wild-type (diagrammed in Supplementary Fig. 1e)[22]. Hermaphrodites from this outcrossed *pot-2* mutant strain were crossed with males containing the *pot-1::mCherry* transgene, and freshly derived F3 *pot-2 -/-* mutant hermaphrodites were observed to possess high levels of POT-1::mCherry foci in all 1- and 2-cell F4 embryos (Supplementary Fig. 1f). Therefore, although telomeres of *pot-2* mutant strains gradually lengthen over many generations of growth, acute loss of POT-2 induces abundant POT-1::mCherry foci in the interphase nuclei of early embryos.

Given the strong effect of deficiency for *pot-2* on early embryonic POT-1::mCherry foci, we introduced an *mNeonGreen*

tag at the endogenous *pot-2* locus using CRISPR/Cas9-mediated genome modification[32]. We confirmed that this version of POT-2 with an N-terminal epitope tag was functional by examining the length of telomeric DNA of an *mNeonGreen::pot-2* strain that had been grown for at least 50 generations. Southern blotting revealed telomeres of wild-type length, in contrast to the elongated telomeres of *pot-2* mutants (Supplementary Fig. 2a). Cytoplasmic mNeonGreen::POT-2 was observed at all embryonic stages and in all adult germ cells (Supplementary Fig. 2b). Like POT-1::mCherry, mNeonGreen::POT-2 was strongly expressed in mature sperm, and embryonic mNeonGreen::POT-2 nuclear foci colocalized perfectly with POT-1::mCherry foci (Supplementary

Fig. 2b). For simplicity, we hereafter refer to nuclear foci composed of *C. elegans* POT-1::mCherry or mNeonGreen::POT-2 proteins as 'Pot1 foci'. mNeonGreen::POT-2 foci were absent from mature -1, -2, and -3 oocytes and from 1-cell interphase embryos and slowly reappeared during embryo development, similar to POT-1::mCherry foci (Supplementary Fig. 2b). However, ample cytoplasmic mNeonGreen::POT-2 levels did not qualitatively vary at these developmental stages (Supplementary Fig. 2b). Therefore, loss of nuclear mNeon-Green::POT-2 foci in 1- and 2-cell wild-type embryos is unlikely to be a consequence of transcriptional or translational regulation of the *pot-2* locus.

**Progeny of *pot-1* and *pot-2* mutants display altered Pot1 foci**. We asked if *pot-2* deficiency in the oocytes of hermaphrodites would alter levels of Pot1 foci in progeny. When males expressing POT-1::mCherry were crossed with *pot-2(tm1400)* mutant her-maphrodites, abundant POT-1::mCherry foci were observed in 1- and 2-cell embryos of *pot-2* heterozygous F1 cross-progeny (diagrammed in Supplementary Fig. 2c). This phenotype per-sisted for six generations, even for F2 lines that were homozygous wild-type for *pot-2*. The number of POT-1::mCherry foci per nucleus in early embryos gradually decreased over six generations but was still statistically increased at generation F7 in comparison to wild-type (Fig. 2a, ANOVA $p < 2E-16$). We confirmed this heritable effect using an independent loss-of-function *pot-2* mutation, *gk162073*. When *pot-2(gk162073)* mutant hermaphro-dites were crossed with wild-type males containing the *pot-1::mCherry* transgene, high levels of POT-1::mCherry foci were observed in all F2 embryos (Supplementary Fig. 2d, ANOVA $p < 10E-14$).

We asked if the induction of POT-1::mCherry foci in the progeny of *pot-2* mutant hermaphrodites was a maternal effect by crossing *pot-2* mutant males with *pot-1::mCherry* hermaphro-dites. This resulted in F1 cross-progeny with abundant POT-1::mCherry foci in all early-stage F2 embryos. This phenotype persisted for at least one additional generation (Fig. 2a). An independent cross of *pot-2* mutant males expressing POT-1::mCherry with hermaphrodites expressing both POT-1::mCherry and mNeonGreen::POT-2 generated F1 cross-progeny with abundant POT-1::mCherry foci that colocalized with mNeon-Green::POT-2 foci throughout embryonic development, including in interphase 1-cell zygotes (Fig. 2b). In contrast, control crosses of *pot-1::mCherry* males with *pot-1::mCherry mNeonGreen::pot-2* hermaphrodites yielded F1 cross-progeny with 1- and 2-cell F2 embryos that lacked Pot1 foci and later-stage embryos with wild-type levels of Pot1 foci (Fig. 2b). Together, these results indicate that gametes of *pot-2* mutants transmit a form of nuclear inheritance that results in abundant telomeric foci composed of POT-1 and, somewhat paradoxically, POT-2 in early embryos for multiple generations.

'Transgenerational inheritance' refers to a phenotype that can be transmitted via an oocyte for at least three generations, whereas 'intergenerational inheritance' is transmitted for only one or two generations[33]. Deficiency for *pot-2* creates gametes that transmit high levels of Pot1 foci for multiple generations. As growth of *pot-1* and *pot-2* mutants results in telomere elongation, we asked if deficiency for *pot-1* would affect levels of Pot1 foci. We crossed males expressing POT-1::mCherry with *pot-1* mutant hermaphrodites. We found that POT-1::mCherry foci were either absent or reduced in number at all stages of embryonic development for two generations of progeny, but levels of POT-1::mCherry foci reverted to wild-type at generation F3 (Fig. 2c, ANOVA $p = 0.101$). We confirmed this intergenerational effect using an independent loss-of-function allele of *pot-1*, *gk177893*,

which induced similarly low levels of POT-1::mCherry foci in all F2 embryos (Supplementary Fig. 2d, ANOVA $p < 10E-14$). We generated an *mNeonGreen::pot-2* strain that was homozygous mutant for *pot-1* and found that mNeonGreen::POT-2 foci were absent from embryos and from most adult germ cell nuclei of this strain, although cytoplasmic mNeonGreen::POT-2 was clearly present (Supplementary Fig. 2e). As gametes derived from *pot-1* or *pot-2* mutants created progeny with low or high levels of Pot1 foci, respectively, we crossed *pot-1::mCherry mNeonGreen::pot-2* males to *pot-2; pot-1* double mutant hermaphrodites and found that F1, F2, and F3 embryos resembled those of *pot-1* mutants, with few foci appearing during development (Fig. 2c). This occurred for descendants of F2 progeny that were homozygous wild-type for *pot-1* and *pot-2*. However, when Pot1 foci appeared at generation F4, their levels were increased in comparison to wild-type and remained slightly elevated for F6 embryos, similar to later-generation progeny of gametes from *pot-2* mutants (Fig. 2c, ANOVA $p = 0.00332$).

Deficiency for *pot-2* promotes the telomerase-independent telomere maintenance pathway ALT[22], which is active in ~10% of tumors[26]. We asked if ALT affects Pot1 foci by crossing *pot-1::mCherry* worms to *trt-1* mutants that had been passaged for over 100 generations under crowded conditions in order to activate the ALT pathway[25]. F1 cross-progeny of ALT strains possessed F2 embryos with wild-type levels of POT-1 and POT-2 foci (Supplementary Fig. 2f), indicating that gametes from animals that maintain their telomeres via the ALT pathway do not create progeny with altered levels of Pot1 foci. We utilized progeny of these crosses to establish a stable *trt-1* mutant strain that expressed POT-1::mCherry and maintained its telomeres by ALT. However, this ALT strain possessed wild-type levels of Pot1 foci (Supplementary Fig. 2g).

**Epigenetic regulation of Pot1 foci by H3K9 methyltransferases**. Gametes of *pot-2* mutants alter levels of Pot1 foci for at least six generations, which is an example of transgenerational epigenetic inheritance that could be associated with factors that can modify gene expression for multiple generations in response to exogen-ous or endogenous cues, including small RNAs, histone mod-ifications, or cytosine methylation[33–37]. We initially tested proteins that affect small RNAs that might guide histone mod-ification at homologous genomic loci. We crossed *pot-1::mCherry* males with hermaphrodites deficient for proteins that promote various small RNA responses, *mut-7*, *mut-14*, *dcr-1*, *hrde-1*, *prg-1*, and *sid-2*. We did not observe significant changes to levels of Pot1 foci in F2 embryos from these crosses (Fig. 3a, ANOVA $p > 0.05$ for each).

Although cytosine methylation does not occur in *C. elegans*[38], telomeres in several species possess features of heterochromatin such as methylation of histone H3K9 or H3K27[39–41]. We therefore asked if Pot1 foci are affected by gametes from mutants with defects in genes that write or erase histone marks[41]. Levels of Pot1 foci were unaffected by gametes deficient for SET-2, which promotes transcription by methylating H3K4[42,43], for SET-4, an H4K20 methyltransferase[44], or for MET-1, which promotes transcription by methylating H3K36[45] (Fig. 3b, ANOVA $p > 0.05$ for each). However, a reduction of Pot1 foci was observed for the F2 embryos of hermaphrodites deficient for SPR-5, which demethylates H3K4me2 in a manner that allows for creation of the H3K9me3 genomic silencing mark (Fig. 3b, ANOVA $p < 10^{-15}$)[46]. Consistently, dramatic reductions in Pot1 foci were observed for F2 progeny of animals deficient for independent alleles of the genome silencing proteins MET-2 and SET-25, which are H3K9 methyltransferases that act redundantly to promote localization of *C. elegans* heterochromatin domains to the nuclear

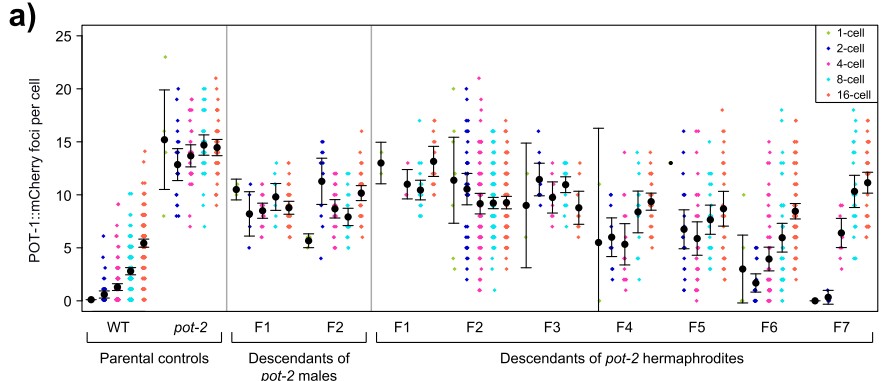

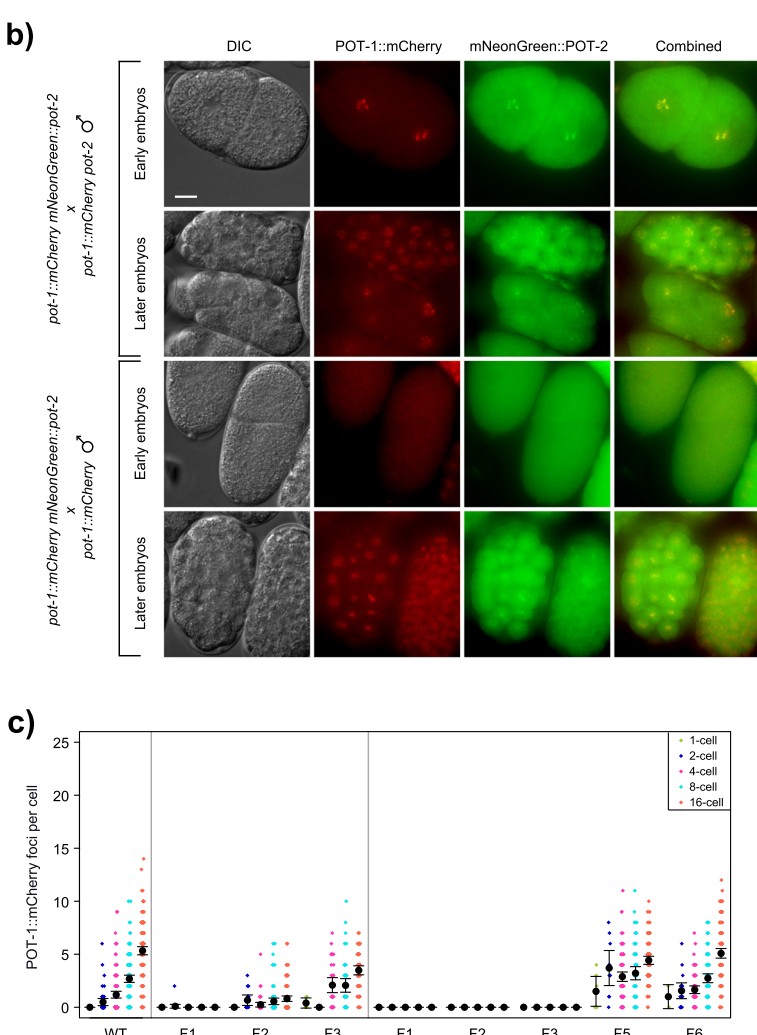

**Fig. 2 *Pot* gene mutations affect Pot1 foci in subsequent generations. a** POT-1::mCherry foci per cell in embryos from P0 parental controls and descendants of either *pot-2* mutant males or hermaphrodites crossed to POT-1::mCherry worms. Dot color indicates stage of embryo. **b** F1 early- and late-stage embryos from crosses between *pot-1::mCherry mNeonGreen::pot-2* hermaphrodites and either *pot-1::mCherry pot-2* males or *pot-1::mCherry* males. Scale bar is 10 μm. **c** POT-1::mCherry foci counts per nucleus in embryos from parental controls and descendants of either *pot-1* or *pot-2; pot-1* mutant males crossed to POT-1::mCherry worms. Error bars are 95% confidence intervals.

periphery[47] (Fig. 3b and Supplementary Fig. 3a, ANOVA $p < 10^{-8}$ for each). In contrast, an increase in Pot1 foci was observed for 1- and 2-cell F2 embryos of hermaphrodites deficient for JMJD-2, a demethylase that targets H3K9 or H3K36 (Fig. 3b, ANOVA $p < 10^{-15}$)[48,49]. We also observed that deficiency for SET-32, an

H3K23 methyltransferase that promotes transgenerational silencing responses to small RNAs, results in progeny with very low levels of Pot1 foci (Fig. 3b, ANOVA $p < 10^{-8}$)[47,50,51].

To further confirm our results, we analyzed a *pot-1::mCherry* strain that was homozygous mutant for the small RNA biogenesis

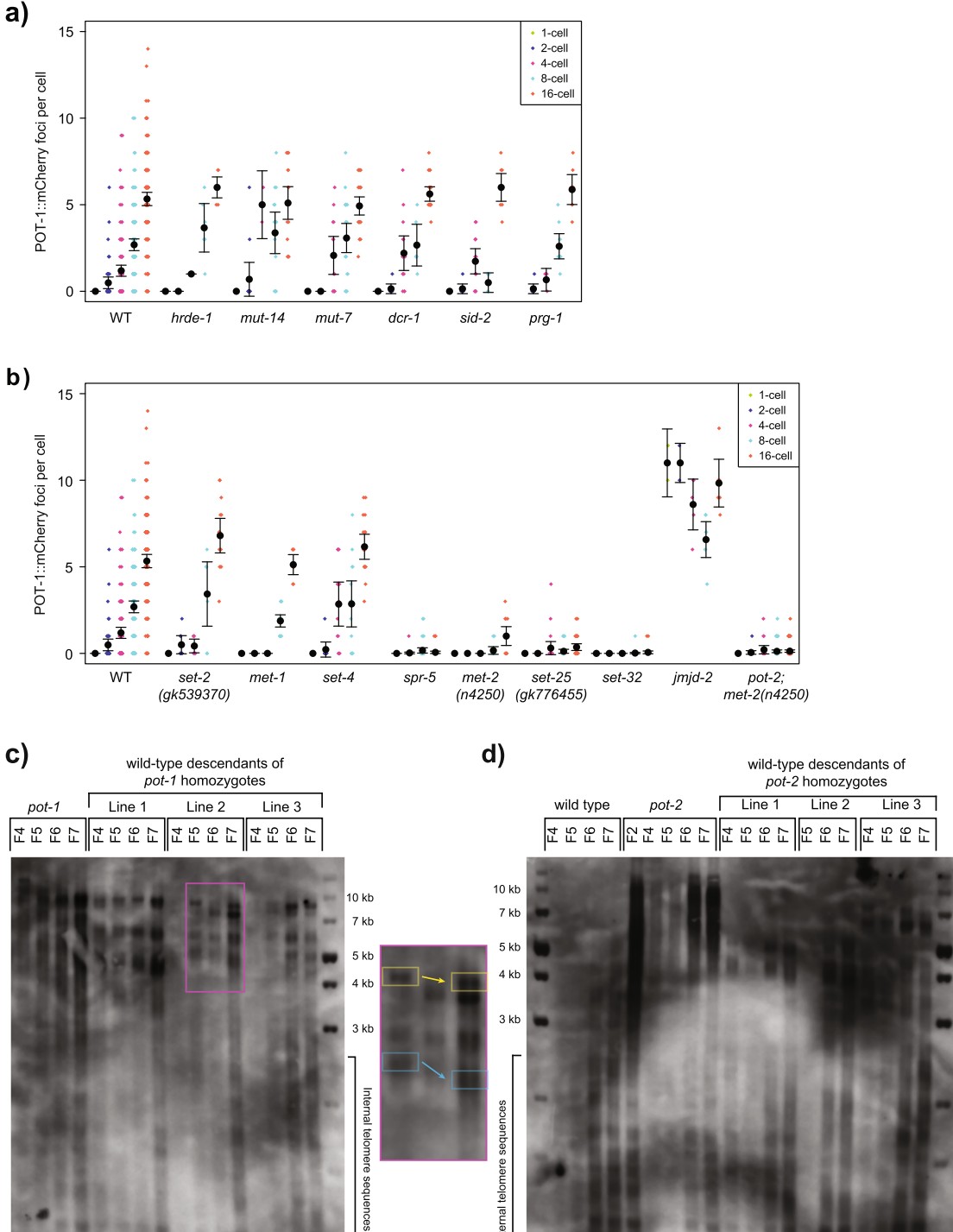

**Fig. 3 Pot1 foci number is epigenetically affected by genes involved in H3K9 methylation. a, b** Quantification of POT-1::mCherry foci in nuclei of F2 embryos derived from crosses of *pot-1::mCherry* males to hermaphrodites mutant for small RNA genes (**a**) or chromatin genes (**b**). **c, d** Southern blot using DNA from the genetically wild-type descendents of crosses between wild-type and *pot-1* (**c**) or *pot-2* (**d**) mutant worms. The portion of the *pot-1* blot indicated by the magenta square is shown at higher detail to the right. Corresponding colored squares show telomere shortening across three generations. Error bars are 95% confidence intervals.

factor *mut-7* and found it had normal levels of Pot1 foci (Supplementary Fig. 3b), consistent with normal levels of Pot1 foci in the F2 progeny of *mut-7* gametes (Fig. 3a). In contrast, we created *pot-1::mCherry* strains that were homozygous mutant for either *met-2* or *set-25* and observed dramatic reductions in levels of Pot1 foci (Supplementary Fig. 3b). We next asked if *met-2* is

epistatic to *pot-2* by creating *pot-2; met-2* double mutant hermaphrodites and observed reduced levels of Pot1 foci in F2 embryos created from crosses with *pot-1::mCherry* males (Fig. 3b, ANOVA $p < 10^{-15}$).

As *pot-1* or *pot-2* mutants possess gametes that give rise to cross-progeny in which levels of Pot1 foci are altered for multiple

generations, we asked if telomere length was perturbed in the progeny of *pot-1* or *pot-2* mutants. We crossed wild-type males to either control wild-type hermaphrodites or to well-outcrossed, early-generation *pot-1* or *pot-2* mutant hermaphrodites that possessed telomeres that are modestly longer than wild-type[22]. F2 cross-progeny that were wild-type for *pot-1* or *pot-2* were identified by PCR. Genomic DNA from the descendants of these genetically wild-type F2 was analyzed by Southern blotting using a probe for *C. elegans* telomere repeats (TTAGGC)$_n$. We found that well-outcrossed *pot-1* or *pot-2* mutant lines had telomeres that were modestly elongated, but when either strain was crossed with wild-type and genetically wild-type F2s were isolated, descendants of gametes generated by either *pot-1* or *pot-2* mutants displayed telomeres that gradually shortened over the course of four generations (Fig. 3c, d). Therefore, gametes of *pot-1* and *pot-2* mutants that alter Pot1 focus levels for multiple generations do not induce the telomere lengthening phenotype that occurs in *pot-1* or *pot-2* mutants[22]. Similarly, telomere length was not consistently elongated in strains deficient for *met-2*, *set-25*, or *set-32* H3 methyltransferases (Supplementary Fig. 3c), which like *pot-1* mutants have gametes that create progeny with dramatic intergenerational reductions in levels of Pot1 foci.

## Discussion

The stability of cytoplasmic mNeonGreen::POT-2 in conjunction with the absence of nuclear POT-2 foci in wild-type 1-cell embryos demonstrates that Pot1 foci are normally dismantled after fertilization (diagrammed in Fig. 4a). Pot1 foci are abundant in wild-type germ cells, decrease as oocytes mature, disappear from 1-cell zygotes, and gradually reappear during the initial cell cycles after fertilization. Although the physiological relevance of this temporal regulation is not clear, the dynamic regulation of telomeres during development is a common theme observed in distinct species. Telomere elongation occurs during blastulation in mouse embryos[52], and telomerase activity spikes during blastulation in bovine embryos[53]. Similarly, components of the shelterin complex are temporally regulated during *Xenopus* development, such that high levels of transcription of telomerase and shelterin proteins were observed in blastula-stage embryos[1,53,54].

Gametes of *pot-1* or *pot-2* mutants altered levels of Pot1 foci for multiple generations (diagrammed in Fig. 4b). Wild-type POT-1 protein must be present in both parents in order for their gametes to create progeny with stable Pot1 foci. In contrast, wild-type POT-2 must be present in both parents in order for their gametes to create progeny with Pot1 foci that are dismantled in 1-cell embryos. Loss of POT-2 in either parent creates germ cells whose progeny contain high levels of embryonic Pot1 foci for six generations, even in the presence of wild-type POT-2. This represents one of the most persistent forms of transgenerational epigenetic inheritance ever documented[33]. However, telomere length was not markedly altered in the progeny of either *pot-1* or *pot-2* mutants. Therefore, although telomerase dysfunction in humans or mice causes transgenerational effects that are associated with inheritance of telomeres of altered lengths, we have identified a form of epigenetic inheritance that modulates the levels of Pot1 foci during development in a manner that is independent of and inconsequential to telomere length. It remains possible that high or low levels of Pot1 foci that are induced by gametes of *pot-2* or *pot-1* mutants, respectively, could affect additional functions of Pot1 that include repression of telomere recombination, repression of resection of the C-rich strand of the telomere, or repression of DNA damage checkpoint activation at telomeres[19,55]. Interestingly, a previous report found that mouse telomeres undergo increased levels of recombination and telomerase-independent extension during the initial divisions of parthenogenetically activated oocytes[56]. However, it is not clear if Pot1 proteins are relevant to this process.

*C. elegans* telomeres have been previously shown to localize to the nuclear periphery in a manner that is lost in *pot-1* mutants but not in *pot-2* mutants or in *met-2; set-25* double mutants[24]. Localization of heterochromatic transgenes and H3K9me-rich chromosome arms to the nuclear periphery is promoted by either MET-2 or SET-25 alone, but is lost in *met-2; set-25* double mutants[47]. In contrast, loss of either MET-2 or SET-25 alone led to dramatic reductions in Pot1 foci. Therefore, the mechanism by which telomeres and other heterochromatic segments of the genome localize within *C. elegans* nuclei is genetically distinct from the mechanism by which POT-1, POT-2, and H3K9 methyltransferases regulate levels of nuclear Pot1 foci. That said,

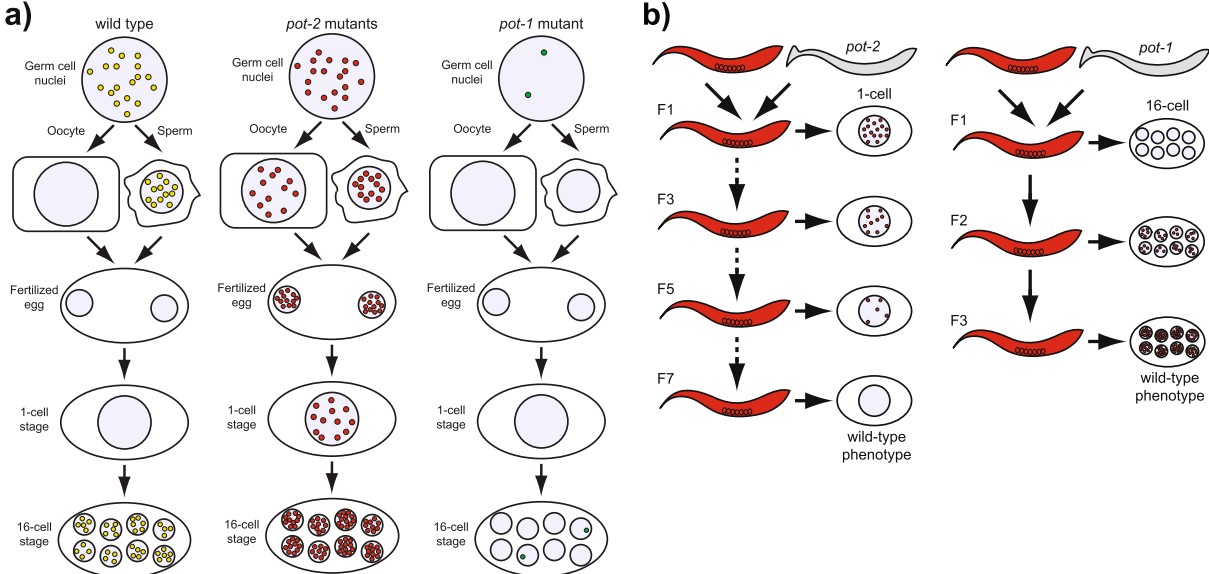

**Fig. 4 Models of Pot1 focus dynamics. a** Descriptive model of the developmental timing of Pot1 foci localization in wild type, *pot-2* mutants, and *pot-1* mutants. Red circles indicate POT-1::mCherry, green circles indicate mNeonGreen::POT-2, and yellow circles indicate colocalization of both transgenic proteins. **b** Descriptive model of the heritable phenotypes of progeny of gametes from *pot-2* or *pot-1* mutants.

it remains possible that gametes from mutants deficient for either *pot-1* or *pot-2* cause a heritable change in telomere positioning or mobility that contributes to or is associated with altered levels of Pot1 foci.

Telomeres can modulate a form of position effect variegation where genes adjacent to telomeres can be epigenetically silenced in a manner that is mitotically heritable and reversible. This epigenetic phenomenon, termed the telomere position effect[57,58], could be related to the mechanism by which gametes of *pot-1* or *pot-2* mutants alter levels of Pot1 foci for several generations. The dominant ability of gametes from *pot-1* or *pot-2* mutants to alter telomeric foci when crossed with wild-type gametes resembles paramutation, where small RNAs produced from a silent locus can act in trans to induce permanent heritable silencing of a distinct locus in the genome[59]. A related silencing process, termed RNAi inheritance, occurs when exogenous small RNAs direct chromatin modifying proteins to induce transient transgenerational silencing of GFP transgenes[35,36,60]. Although we did not observe an effect of small RNA biogenesis factors on Pot1 foci, loss of two H3K9 methyltransferases, MET-2 and SET-25, or loss of the H3K23 methyltransferase SET-32 mimicked the low levels of Pot1 foci observed in response to deficiency for *pot-1* until the F2 generation. We did not test for an effect of these methyltransferase mutations in later generations. However, previous reports have observed that the H3K9 methyltransferases MET-2 and SET-25 can mediate forms of transgenerational epigenetic inheritance where transgene silencing is modulated by environmental or endogenous cues[61,62]. Support for a role for histone methylation in creating heterochromatic Pot1 foci is provided by strains deficient for the H3K9 demethylase *jmjd-2*[49], which created gametes that induced high levels of Pot1 foci for multiple generations, similar to gametes of *pot-2* mutants. As small RNA biogenesis factors did not influence Pot1 focus formation at telomeres, recruitment of histone H3 methyltransferases to telomeres might be carried out by a sequence-specific telomere binding protein like POT-1.

Telomeric repeats in *C. elegans* and humans are enriched in H3K9me2[29], while human subtelomeres are enriched in H3K9me3[63,64]. We found that telomeric foci were perturbed by mutations in either *met-2* or *set-25*, two histone methyltransferases that create H3K9me2 and H3K9me3, respectively[47,65]. In addition, the loss of the H3K4 demethylase *spr-5* reduced telomeric Pot1 foci. While SPR-5 may demethylate H3K9[66], its main activity is demethylation of H3K4, which promotes H3K9 methylation[46]. Interestingly, the drop in Pot1 foci we observed during oogenesis and early embryonic development coincides with decreased levels of nuclear histone methylation. H3K9me2 is high in pachytene germ cells but is lost in mature oocytes at diakinesis[65]. Furthermore, H3K9me1, me2, and me3 levels were previously demonstrated to be low in 1-cell embryos and then to rapidly increase during early embryogenesis[67]. The developmental dynamics of global levels of H3K9 methylation could be related to the mechanism by which Pot1 focus formation is transiently eliminated in wild-type embryos.

Although the absence of MET-2 and SET-25 together abolishes all germline H3K9 methylation[47], we found that these proteins are independently required to promote the stability of Pot1 foci. A third histone methyltransferase that is required for Pot1 focus formation, SET-32, methylates H3K23, which is a heterochromatic mark that has been recently identified in *C. elegans*[51,68]. SET-25 and SET-32 have previously been shown to respond to small RNAs by establishing silent chromosome domains that are then maintained for multiple generations[69]. Loss of MET-2 affects fertility in a manner that becomes apparent only after growth for multiple generations[70], which suggests transmission of a heritable epigenetic factor by gametes. As the stability of Pot1

foci strongly depends on H3K9 and H3K23 methyltransferases that have previously been demonstrated to promote heterochromatin formation at internal segments of the genome, we suggest that Pot1 foci correspond to silent chromosome domains at chromosome termini.

We addressed high levels of Pot1 foci in the descendants of gametes from *pot-2* mutants by examining the progeny of *pot-2; pot-1* and *pot-2; met-2* double mutants, which did not display Pot1 foci in embryos. This implies that the presence of POT-1 and H3K9 methylation at telomeres of gametes from *pot-2* mutants triggers high levels of telomeric foci for multiple generations. The precise mechanistic relationship between POT-1, MET-2, SET-25, SET-32, and H3K9 or H3K23 methylation and between POT-2, JMJD-2, and H3K9 demethylation in the context of transgenerational epigenetic inheritance of Pot1 foci are intriguing questions that remain to be explored experimentally.

Jean Baptiste Lamarck and Charles Darwin postulated that environmental stimuli might induce the transgenerational epigenetic inheritance of traits that promote fitness in future generations[71]. Although we do not understand the functional significance of altered levels of Pot1 foci, previous analysis of the genomes of 152 wild *C. elegans* strains revealed that 12 possessed a putative inactivating mutation in *pot-2* that was associated with long telomeres, suggesting that either POT-2 or telomere length could contribute to fitness in the wild[72].

We conclude that Pot1 proteins regulate persistent forms of intergenerational and transgenerational epigenetic inheritance that influence developmental expression of Pot1 foci at telomeres. Regulation of telomeres has been tied to aging and cancer, and mutations of the Pot1 locus in human families are associated with an increase in the incidence of cancer[30,31,73,74]. If transgenerational epigenetic inheritance of Pot1 foci is relevant to human health, then Pot1 mutations segregating in human pedigrees might induce traits in future generations that could affect family members that lack the Pot1 mutation[30].

## Methods

**Strains**. All strains were cultured at 20 °C on Nematode Growth Medium (NGM) plates seeded with *Escherichia coli* OP50, unless otherwise indicated. Strains used include Bristol N2 wild-type, YA1197 *ypSi2 (pdaz-1::pot-1::mCherry::ttb-2 3′UTR)*, VC20674 *pot-1(gk177893)*, YA1022 *pot-1(tm1620) III*, YA997 *trt-1(ok410) I*, BR3417 *spr-5(by134) I*, YA1024 *pot-2(tm1400) II*, VC40055 *pot-2(gk162073)*, YA1249 *pot-2(ypSi3 [mNeonGreen::pot-2]) II*, CB444 *unc-52(e444) II*, NL1820 *mut-7(pk720) III*, YY470 *dcr-1(mg375) III*, NL1838 *mut-14(pk738) V*, HC196 *sid-2 (qt42)*, YY538 *hrde-1(tm1200)*, MT16973 *met-1(n4589)*, MT14851 *set-2(n4589)*, VC40257 *set-2(gk539370)*, MT14911 *set-4(n4600)*, MT17463 *set-25(n5021)*, VC40568 *set-25(gk697056)*, VC40718 *set-25(gk776455)*, SX922 *prg-1(n4357)*, MT13293 *met-2(n4256)*, VC40243 *met-2(gk531543)*, VC40995 *met-2(gk919134)*, VC967 *set-32(ok1457)*, and VC20722 *jmjd-2(gk383243)*.

Passaging of the YA1197 strain in the presence of males for dozens of generations sometimes results in silencing of the transgene, despite *ypSi2* being a single-copy insertion[22]. When transgene silencing occurred, we thawed a frozen stock of YA1197 hermaphrodites.

**Generation of fresh *pot-2* homozygous mutants from balanced stocks**. The *pot-2* locus and the *pot-1::mCherry* transgene both reside on Chromosome II. *pot-2* is very close to *unc-52*, while *pot-1::mCherry* is very close to *rol-6*. We created heterozygous *pot-2(tm1400)* male stock by crossing *pot-2(tm1400)* hermaphrodites with an *unc-52 − / +* males, crossing single F1 males with *unc-52* mutant hermaphrodites, choosing crosses that yielded Unc F1 males, and crossing their non-Unc male siblings with *unc-52* hermaphrodites. The *pot-2(tm1400) / unc-52* males were repeatedly crossed with *unc-52* hermaphrodites. A *rol-6 unc-52* double mutant strain was used to create *pot-1::mCherry, unc-52* double mutants. *pot-2(tm1400) / unc-52* males were crossed with *rol-6, unc-52* double mutants, non-Unc F1 males were crossed with *pot-1::mCherry, unc-52* hermaphrodites to create *pot-2 − / +* F1 hermaphrodites possessing a single copy of *pot-1::mCherry* (Supplementary Fig. 1e). *pot-2* heterozygotes possessing two copies of *pot-1::mCherry* were obtained by selecting F2 worms that lacked Rol F3 but had some Unc F3 progeny. Most F3 embryos produced by these F2 worms did not display abundant early foci. However, a few, most likely the *pot-2* homozygous mutants, did display early foci. We also studied F3 adult progeny that were *pot-1::mCherry; pot-2 − / −* homozygotes and found high levels of POT-1::mCherry foci in all 1- and 2-cell F4

embryos. *pot-2* mutants that are created from well-outcrossed *pot-2* mutant stocks have normal telomere lengths at generation F4, which slowly increase in length over the course of 16 generations[22].

**Generation of *mNeonGreen::pot-2* transgenic strain**. DNA constructs for the *mNeonGreen::pot-2* transgene were generated based on the previously described SapTrap protocol[75]. CRISPR-based insertion into the genome was accomplished using previously described protocols[32]. Sanger sequencing of the *mNeonGreen* tag and neighboring genomic region confirmed the *mNeonGreen* insertion.

We generated epitope tags at the endogenous *pot-1* locus through similar means. Though we were successful in generating both N- and C-terminal *mKate* tags of *pot-1*, we were unable to visualize the transgenes. This may be due to weak expression from the endogenous *pot-1* locus, as expression of *pot-2* is considerably higher than *pot-1* based on RNAseq data from the modENCODE project[76].

**Generation of *trt-1; pot-1::mCherry mNeonGreen::pot-2* ALT lines**. *trt-1* mutant worms were passaged for over 100 generations by chunking in order to obtain lines that maintained telomeres by ALT. Individual L4 ALT hermaphrodites were mated to males homozygous for *pot-1::mCherry* and *mNeonGreen::pot-2*. Singled F2 homozygotes containing both the fluorescent tags and the *trt-1* mutation were identified by PCR. These lines were then passaged by chunking for another 20 generations to ensure continued activation of ALT.

**Crosses**. To perform crosses with strains containing *pot-1::mCherry* or *mNeon-Green::pot-2*, 24 males and 10 hermaphrodites were placed on a single NGM plate. Plates were incubated at 15 °C for 3 days to obtain F1 cross-progeny embryos from hermaphrodites that were all mated.

**Imaging**. Whole worms and embryos were mounted on 2% agarose pads and imaged live using a DeltaVision Elite microscope (Applied Precision) with a 60x Silicon oil objective lens. For germline imaging, immobilization was assisted by transferring worms into 5 µL drops of 0.2 mM levamisole on agarose pads. For embryo imaging, embryos were extruded from adults in M9 buffer and embryos were transferred onto agarose pads using a mouth pipet.

**Statistics and reproducibility**. Images were analyzed using ImageJ software. Foci were counted manually from fluorescent images, while embryo stage was determined from DIC images.

Statistical calculations were performed using R software. Focus numbers were compared using either the Wilcoxon rank-sum test or 2-way ANOVA, depending on the number of variables tested simultaneously. When making multiple simultaneous statistical comparisons, *p*-values were adjusted using the Holm-Bonferroni method. All error bars displayed represent 95% confidence intervals. Sample sizes for imaging experiments are listed in Supplementary Data 1.

## Data availability

Image datasets used for the generation of figure graphs can be found online at Dryad, https://doi.org/10.5061/dryad.mkkwh70xz[77].

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

## Acknowledgements
We thank Felix Peng for initial observations of high levels of POT-1::mCherry foci in early embryos of *pot-2* mutants. Dan Dickinson kindly provided plasmids used in the generation of the *mNeonGreen::pot-2* targeting construct. We thank Bob Goldstein and Dan Dickinson for advice regarding CRISPR-mediated genome modifications. We are indebted to Amy Maddox for training on and access to her DeltaVision microscope. P.M. is the William Burwell Harrison Scholar and supported by NSF CAREER award #1652512. S.A. is supported by NIH grant R01GM135470.

## Author contributions
E.L.S. and M.D. performed the experiments. E.L.S analyzed the data. P.M. and S.A. supervised the experiments. S.A. and E.L.S. wrote the manuscript with input from M.D. and P.M.

## Competing interests
The authors declare no competing interests.

## Additional information
**Supplementary information** Supplementary information is available for this paper at https://doi.org/10.1038/s42003-020-01624-7.

