## [Peer Review File · Communications Biology]

Reviewers' Comments:

Reviewer #1:

Remarks to the Author:

The authors describe an interesting and unexpected example of transgenerational epigenetic inheritance whereby inactivation of either *pot-2* or *pot-1* in the adult germline results in a transgenerational epigenetically inherited phenotype whereby POT foci can either not be dissembled in embryos or do not assemble, respectively, for many generations. The authors correctly point out that this is one of the longest (6 generation) epigenetic memories described. The mechanism of inheritance is very unclear and the physiological relevance is unknown (the phenotype is induced by mutations; no additional phenotypes are observed beyond the altered foci). Nonetheless, this is an interesting discovery that may lead to the elucidation of a novel mechanism of epigenetic inheritance.

However, the current manuscript does need to be substantially revised before it can be published because it is full of over-interpretations of the data, inaccuracies, and mistakes. The most important point is that the authors go too far when seeming to rule out standard epigenetic mechanisms (small RNAs, histone marks) based on negative results in the very small subset of genes which they test, and they are very careless in their descriptions of what they tested (repeatedly referring to heterochromatin, H3K9 and genomic silencing when the only (!) gene they tested was the H3K4 demethylase *spr-5*). If they really want to make this claim in such a generalised way they have to test more pathways/enzymes. Alternatively they need to state that they actually haven't tested most known mechanisms and to be much more precise and careful in the description of what they have - and have not - tested and shown.

Given the number of statements not supported by experiments and the number of mistakes, it seems that the senior author may not have properly checked the text or figures of the manuscript before submission.

Specific comments:

The abstract is complicated and could be re-written to better communicate the main finding rather than simply listing the experiments and results. It also currently over-interprets the results.

The authors should be clear in the text that the mechanism of inheritance is unknown and the physiological relevance is also unclear.

The authors write that this is a new form of epigenetic inheritance because it does not involve small RNAs or chromatin modifications. However this is an overstatement given the actual experimental data. Many known genes and mechanisms have not been excluded so this sentence needs to be removed.

They need to either test requirement of more chromatin and small RNA pathways for inheritance of *pot-1* and *pot-2* mutant effects on the foci OR tone down the language to state that the inheritance may involve some untested known mechanisms.

Moreover they don't actually test if small RNA/chromatin pathways affect the inheritance, they test whether they influence the POT foci when inactivated alone. Again they should either test for an effect on the inheritance OR tone down the language to make this clear.

"These data indicate that disrupting small RNAs or H3K9-mediated genomic

silencing does not perturb levels of Pot1 foci.” – did not test H3K9. Only tested H3K4 demethylase spr-5, which leads to accumulation of H3K4me2. Some link has been shown between H3K4 (specifically spr-5) and H3K9 in paper by Greer 2014; but this paper is not even cited.

Likewise: “However, both Paramutation and Position Effect Variegation are genomic silencing processes, and we found that defects in heterochromatin formation...”

They did not test heterochromatin formation. They really should test a wider range of marks to make this point.

In Methods, proper strain names please, not only genotypes.

Some of the results described in the text do not appear to be shown in the figures:

Fig 2a: for the male cross only 1 generation is shown, and not an ‘additional 2 generations’ as indicated in the text.

Show the data for these two statements: ‘We observed high levels of POT-1::mCherry foci in all 1- and 2-cell embryos derived from progeny of pot-2 -/- animals whose mothers were pot-2 +/- heterozygotes. We also found that heterozygosity for pot-2 does not result in abundant POT-1::mCherry foci in early embryos’.

Show the data for ‘and wild-type levels of Pot1 foci were observed at later stages of development’

Fig 2B: Show the data for more generations for the pot-1 mutant descendants to see whether levels revert to wild type levels as they did for pot-1;pot-2.

The data for this statement is not shown: ‘although cytoplasmic mNeonGreen::POT-2 was unaffected in the progeny of pot-2; pot-1 double mutant gametes (Fig 2B)’

Fig S3E+F: the ‘gradual shortening’ of the telomeres is not obvious in the blots. It looks like in later generations the telomeres become longer instead. It would be good to use arrows or other graphical aid to explain the changes in the blot. Also, within the pot-2 control there seems to be a lot of variation in telomere length between F2 and F6? Is there a biological repeat for this important experiment? As a strong argument is built on the telomere dynamics in the mutants, the blots should be more clearly explained and probably be included in the main figures.

‘The ability of telomeres of pot-1 and pot-2 mutant gametes to transform developmental expression of Pot1 foci’. I don’t think there is evidence to exclude that it could be another characteristic of the mutants other than the telomeres that causes the transgenerational effect.

‘We conclude that C. elegans POT-1 functions at telomeres’. I did not see evidence to exclude that POT-1 might act elsewhere.

What is the n number of animals/oocytes included in each experiment?

Typos (there are lots)

“human families causes in transgenerational shortening”

Throughout the text many things capitalised (e.g. Transgenerational Epigenetic Inheritance, Paramutation, Telomere Position Effect). Don't capitalise these things.

"oocytes" page 4

Formatting italics for consistency in Methods "Generation of mNeonGreen::pot-2 transgenic strain"

Fig S3E left panel. Line 3,3,3. Should be 1,2,3.

In Methods, 5 microlitres expressed as 5 μ L. Please use greek symbols.

C. elegans nomenclature should be used consistently: 'POT-1 (foci)' instead of 'Pot1 (foci)'

Reviewer #2:

Remarks to the Author:

In this study, Lister-Shimauchi et al developed transgenic C. elegans lines that express fluorescent proteins that are fused to POT-1 and POT-2, which are telomere binding proteins regulating telomere length, and successfully visualized localization of POT-1 and POT-2 in embryos. By using this system, the authors demonstrated that POT-1::mCherry and mNeonGreen::POT-2 fusion proteins make foci in adult germ cells that vanish in 1-cell embryo, and gradually accumulate again during embryonic development. The authors also found that, in embryos from pot-2 mutant gametes, POT-1 and POT-2 fusion proteins form numerous foci throughout embryonic development, and very interestingly, this unexpected foci formation phenotype is epigenetically inherited for 6 generations, which is one of the longest types of transgenerational epigenetic inheritance ever documented. In addition, the authors also found that deficiency for POT-1 in gametes results in reduced foci number of POT-1::mCherry in all stages of embryos, and this phenotype is also epigenetically inherited for multiple generations. This study reveals a novel form of transgenerational epigenetic inheritance related to telomeres. I believe that this study provides important new insights into transgenerational epigenetic inheritance. Overall, experiments are well designed and the impression of this study is very positive. One concern is that transgene expression could alter expression and localization of endogenous genes, and fusion proteins could exhibit unexpected functions. Therefore, the authors should investigate the expression, localization, and function of fusion proteins as well as the effects of transgene expression on endogenous proteins more carefully. Especially, the authors need to show that POT-1::mCherry and mNeonGreen::POT-2 foci indeed are actually at telomeres, and that homozygous mNeonGreen::POT-2 animals do not exhibit the phenotype found in pot-2 mutant animals. In addition, if the authors could demonstrate the foci formation of endogenous POT-1 in early embryos from pot-2 mutant gametes, it would significantly improve the quality of this study.

Minor comments

1. A breeding scheme of each experiment would help readers to understand the manuscript.
2. Some of the results described in the manuscript are missing.
 - 1) Page4, line5: high levels of POT-1::mCherry foci in all 1- and 2-cell embryos derived from progeny of pot-2 animals whose mothers were pot-2 +/- heterozygous
 - 2) Page4, line7: heterozygosity for pot-2 does not result in abundant POT-1::mCherry foci in early embryos.
 - 3) Page5, line19: wild-type levels of Pot1 foci were observed at later stages of development.

Reviewer #3:

Remarks to the Author:

Review of manuscript COMMSBIO-19-0412-T

Key results

The nematode *C. elegans* is an excellent system for studying transgenerational epigenetic inheritance mechanisms, including small-RNA-induced epigenetic silencing and paramutation. In this work, the authors report on a novel case of transgenerational memory in worms, which affects the nuclear foci formed by the POT-1 and POT-2 telomere binding proteins. The work is based on the peculiar observation that single copy POT-1::mCherry has the ability to form developmentally-regulated telomere foci in vivo, which are prominent in the adult germ cells, disappear in 1-cell embryos and are progressively reconstituted during embryonic development. Also POT-2 (tagged with mNeonGreen) has the ability to form foci in vivo, which behave similarly to POT-1 foci. The authors show that mutants deficient for the POT-2 protein lost the ability to regulate the development of such foci, which then persist throughout their life cycle. Interestingly, gametes from *pot-2* mutant strains transmit the ability to form POT-1 (and POT-2) foci irrespective of developmental stage. Formation of these foci requires intact POT-1 function. Once exposed to POT-2 deficiency, the memory of deregulated formation of POT1/2 foci acquired from *pot-2* mutant gametes persists over many generations, even in the presence of functional POT-2. This transgenerational effect does not appear to be responsive to perturbation in the RNAi machinery or to modulation of chromatin structure.

Validity, originality and significance: Although there is no clear mechanistic explanation for the transgenerational inheritance of the *pot-2* -dependent deregulation of POT-1 and POT-2 foci, the work has a robust genetic outline and describes a novel aspect of the complex epigenetic regulation of telomeres. This is of interest not only to specialists in chromosome maintenance and stability, as it uncovers a novel example of acquired epigenetic memory, a phenomenon that persists over many generations, long after the exposure to the initial perturbation.

Data & methodology

The experimental outline is sound. Specific comments are provided below.

Appropriate use of statistics and treatment of uncertainties

Statistical analyses should be included for figure 2A and 2B. All other analyses are appropriate.

Conclusions

Further studies will be required to shed light on the mechanisms governing the described transgenerational epigenetic inheritance of the developmentally regulated formation of POT-1 and POT-2 nuclear foci. As pointed out below, the authors should integrate the discussion of the regulation of POT-1 and POT-2 foci formation with other known aspects of POT-1 and POT-2 biology, such as their ability to modulate telomere recombination and telomere length.

Suggested improvements

1) Previous work from the same group (Shtessel et al., 2013 DOI: 10.1534/g3.112.004440) has shown that POT-1 forms foci at telomeres. This is particularly clear in pachytene nuclei, in which an average of 12 POT-1::mCherry foci /nucleus are visible, likely identifying the telomeres of six paired homologous chromosomes, while 19 foci in average are visible in mitotic nuclei, compared to the expected 24 foci. To improve clarity, it would be appropriate to include here a brief explanation of what the number of expected foci is, relative to the number of chromosomes. This elucidation is relevant, as there is no actual proof that the POT1::mCherry foci indeed colocalize with telomeres, besides the fact that the number of foci is compatible with the number of telomere ends. No colocalization with a telomere probe is shown.

2) The authors previously suggested that telomere clustering may account at least in part for the deviation in comparison to the number of expected foci in wild type cells. Work from the Gasser group (Ferreira et al., 2013 doi.org/10.1083/jcb.201307181) suggested that telomeres are enriched at the

nuclear periphery and tend to be clustered in strains which activated the ALT pathway. Peripheral localization of telomeres appears to be unaffected in *pot-2* mutants, while POT1 is capable of mediating telomere positioning at the nuclear periphery. The authors should discuss the possibility that the number of observed foci may reflect the state of telomere clustering or their positioning within the nucleus. Could POT2-deficiency induce a heritable change in telomere anchoring or mobility that accounts for the increase in POT1 foci observed in later generations? Can they comment as to whether the disappearance of POT1::mCherry and mNeonGreen::POT-2 foci in *pot-1* mutants is related to loss of POT-1-mediated telomere anchoring at the nuclear periphery?

3) The quality of the TRF in Supplementary Fig. 3E and F is not optimal for conclusion that telomere shortening actually occurs in the 4 generations explored. The authors should include labels in the TRF lanes to help locate mean telomere length. They should also point out if telomere heterogeneity is observed in *pot-1* and *pot-2* derived strains.

4) POT-2 deficiency has been shown to affect the amount of telomeric recombination and C-circles, especially at later generations. Is it possible that POT-2 deficiency may exert a dominant effect on telomere recombination, which persists in late generation descendants of *pot2*- gametes?

Additional points

- Figure 2A and 2B: statistical analyses for multiple comparisons between classes should be provided
- The labeling of the *pot-2* and *pot-1* derived lines in FIG.S3E and S3F is confusing.
- Genetic crosses in Fig. S1P: check the labeling of the markers in the F1 hermaphrodites

Point by point responses for Lister et al., "Gametes deficient for Pot1 telomere binding proteins alter levels of telomeric foci for multiple generations"

Reviewer #1 (Remarks to the Author):

Comment 1:

The authors describe an interesting and unexpected example of transgenerational epigenetic inheritance whereby inactivation of either *pot-2* or *pot-1* in the adult germline results in a transgenerational epigenetically inherited phenotype whereby Pot1 foci can either not be disassembled in embryos or do not assemble, respectively, for many generations. The authors correctly point out that this is one of the longest (6 generation) epigenetic memories described. The mechanism of inheritance is very unclear and the physiological relevance is unknown (the phenotype is induced by mutations; no additional phenotypes are observed beyond the altered foci). Nonetheless, this is an interesting discovery that may lead to the elucidation of a novel mechanism of epigenetic inheritance.

However, the current manuscript does need to be substantially revised before it can be published because it is full of over-interpretations of the data, inaccuracies, and mistakes. The most important point is that the authors go too far when seeming to rule out standard epigenetic mechanisms (small RNAs, histone marks) based on negative results in the very small subset of genes which they test, and they are very careless in their descriptions of what they tested (repeatedly referring to heterochromatin, H3K9 and genomic silencing when the only (!) gene they tested was the H3K4 demethylase *spr-5*). If they really want to make this claim in such a generalised way they have to test more pathways/enzymes. Alternatively they need to state that they actually haven't tested most known mechanisms and to be much more precise and careful in the description of what they have - and have not - tested and shown. Given the number of statements not supported by experiments and the number of mistakes, it seems that the senior author may not have properly checked the text or figures of the manuscript before submission.

Response 1:

Thank you for these comments. The primary reason for our previous enthusiasm about our negative results with small RNA factors is that we have a distinct manuscript where we show that these factors promote telomere stability in the absence of telomerase. These factors can act at telomeres but do not affect Pot1 foci. At the reviewers suggestion, we tested mutations in additional genes that are known to or might affect chromatin modifications and found that three genome silencing factors, *met-2*, *set-25* and *set-32*, two of which methylate H3K9, are required for assembly of Pot1 foci. Although SET-25 and SET-32 can respond to small RNAs, our data indicate that small RNAs do not affect Pot1 focus levels. We also tested *jmjd-2*, an H3K9 demethylase, and found it is required for disassembly of Pot1 foci. Although we have not tested all known epigenetic pathways, we have broadened the number of epigenetic factors tested and obtained positive and negative results that suggest a tantalizing model that Pot1 foci may be silent chromatin domains decorated with H3K9 methylation.

Comment 2:

Specific comments:

The abstract is complicated and could be re-written to better communicate the main finding rather than simply listing the experiments and results. It also currently over-interprets the results

Response 2:

We have carefully edited the abstract for clarity and attempt to conservatively interpret the significance of our results.

Comment 3:

The authors should be clear in the text that the mechanism of inheritance is unknown and the physiological relevance is also unclear.

Response 3:

We have incorporated new results using histone methyltransferase and demethylase mutations into the paper, which provide insight into the mechanism of inheritance. Unfortunately, the physiological relevance remains unknown, and we try to clearly convey this.

Comment 4:

The authors write that this is a new form of epigenetic inheritance because it does not involve small RNAs or chromatin modifications. However this is an overstatement given the actual experimental data. Many known genes and mechanisms have not been excluded so this sentence needs to be removed.

They need to either test requirement of more chromatin and small RNA pathways for inheritance of *pot-1* and *pot-2* mutant effects on the foci OR tone down the language to state that the inheritance may involve some untested known mechanisms.

Response 4:

Happily we have identified multiple chromatin genes that affect Pot1 foci, two of which are known to play roles in transgenerational inheritance in response to small RNAs. We continue to observe negative results when siRNA biogenesis is perturbed.

Comment 5:

Moreover they don't actually test if small RNA/chromatin pathways affect the inheritance, they test whether they influence the Pot1 foci when inactivated alone. Again they should either test for an effect on the inheritance OR tone down the language to make this clear.

Response 5:

Experimentally separating inheritance from establishment of altered levels of Pot1 foci is a terrific idea. We have done this for both *pot-1* and *pot-2* progeny and observed that both mutations are required for initiation but not inheritance. Our preliminary results indicate that *met-2* and *set-25* are also required for initiation but not inheritance of low levels of Pot1 foci,

although we have not yet quantified this. We have tried to make the wording of the manuscript consistent with the current data.

Comment 6:

“These data indicate that disrupting small RNAs or H3K9-mediated genomic silencing does not perturb levels of Pot1 foci.” – did not test H3K9. Only tested H3K4 demethylase *spr-5*, which leads to accumulation of H3K4me2. Some link has been shown between H3K4 (specifically *spr-5*) and H3K9 in paper by Greer 2014; but this paper is not even cited.

Response 6:

We retested a validated *spr-5* mutant line and found that *spr-5* mutant gametes phenocopy the low levels of Pot1 foci observed in response to deficiency for *pot-1*. Furthermore, we tested mutations for two H3K9 methyltransferases MET-2 and SET-25 and found that these also phenocopied deficiency for *pot-1*. We now discuss the activity of *spr-5*, cite the Greer 2014 paper, and directly test two H3K9 methyltransferases.

Comment 7:

Likewise: “However, both Paramutation and Position Effect Variegation are genomic silencing processes, and we found that defects in heterochromatin formation...” They did not test heterochromatin formation. They really should test a wider range of marks to make this point.

Response 7:

We previously observed negative results for RNAi knockdown for a larger number of chromatin proteins, including *met-2*, *set-25*, and *set-32* (and also for *pot-1* and *pot-2*), even though positive control RNAi clones worked robustly. As RNAi knockdown can yield false-negative results, we performed genetic tests using loss-of-function mutations in two proteins that promote transcription, SET-2 and MET-1, as well as three proteins that promote heterochromatin formation, MET-2, SET-25, and SET-32, and one protein that dismantles H3K9me heterochromatin marks, JMJD-2. This arguably remains a limited set of proteins that affect chromatin, but the positive results with MET-2, SET-25, SET-32 and JMJD-2 complement the POT-1 and POT-2 data that we present.

Comment 8:

In Methods, proper strain names please, not only genotypes.

Response 8:

We have added strain names.

Comment 9:

Some of the results described in the text do not appear to be shown in the figures: Fig 2a: for the male cross only 1 generation is shown, and not an ‘additional 2 generations’ as indicated in the text.

Response 9:

Thank you for pointing this out. We have corrected the text.

Comment 10:

Show the data for these two statements: 'We observed high levels of POT-1::mCherry foci in all 1- and 2-cell embryos derived from progeny of *pot-2* $-/-$ animals whose mothers were *pot-2* $+/-$ heterozygotes. We also found that heterozygosity for *pot-2* does not result in abundant POT-1::mCherry foci in early embryos'.

Response 10:

We have added a panel for the first statement (Supplementary Fig. 1F). The indicated paragraph was shortened slightly due to new data being introduced throughout, so the second statement no longer appears for simplicity's sake. It could be added back in with a data panel if necessary.

Comment 11:

Show the data for 'and wild-type levels of Pot1 foci were observed at later stages of development'

Response 11:

We have incorporated data panels showing later stage embryos for both control and *pot-2* mutant crosses (Fig. 2B).

Comment 12:

Fig 2B: Show the data for more generations for the *pot-1* mutant descendants to see whether levels revert to wild type levels as they did for *pot-1;pot-2*.

Response 12:

Based on ANOVA, foci levels for the descendants of *pot-1* mutants revert to wild-type by generation F3. We have included statistics in the text.

Comment 13:

The data for this statement is not shown: 'although cytoplasmic mNeonGreen::POT-2 was unaffected in the progeny of *pot-2*; *pot-1* double mutant gametes (Fig 2B)'

Response 13:

We now clarify that qualitative evaluation of cytoplasmic mNeonGreen::POT-2 suggested that this was unaffected in progeny of *pot-2*; *pot-1* double mutants.

Comment 14:

Fig S3E+F: the 'gradual shortening' of the telomeres is not obvious in the blots. It looks like in later generations the telomeres become longer instead. It would be good to use arrows or other

graphical aid to explain the changes in the blot. Also, within the *pot-2* control there seems to be a lot of variation in telomere length between F2 and F6? Is there a biological repeat for this important experiment? As a strong argument is built on the telomere dynamics in the mutants, the blots should be more clearly explained and probably be included in the main figures.

Response 14:

We expected to see a modest increase in telomere length for both *pot-2* and *pot-1* mutant controls over the course of a few generations, and our results were consistent with previously published data (Shtessel 2013). We performed 3 biological replicates for the crosses, but did not perform replicates for the controls as they were consistent with previously published results for *pot-1* and *pot-2* mutants. We hope that our results are more clear based on a graphical explanation of the blot, which is now included in the figure (Fig. 3E).

Comment 15:

'The ability of telomeres of *pot-1* and *pot-2* mutant gametes to transform developmental expression of Pot1 foci'. I don't think there is evidence to exclude that it could be another characteristic of the mutants other than the telomeres that causes the transgenerational effect.

Response 15:

Thank you for this comment. Our assumption is that the telomeres of these gametes are responsible, but it is formally possible that an alternative cue could induce the transgenerational effects. We therefore softened this statement by stating "the ability of *pot-1* and *pot-2* mutant gametes to transform levels of Pot1 foci". The statement no longer specifies that telomeres are responsible.

Comment 16:

'We conclude that *C. elegans* POT-1 functions at telomeres'. I did not see evidence to exclude that POT-1 might act elsewhere.

Response 16:

This statement has been softened to "We conclude that Pot1 proteins regulate a persistent form of transgenerational epigenetic inheritance that influences developmental expression of Pot1 foci at telomeres". The statement no longer refers to the location of Pot1 action.

Comment 17:

What is the n number of animals/oocytes included in each experiment?

Response 17:

A supplemental table was created to give sample sizes for each experiment. Generally, we used 2-10 oocytes per animal and at least 10 animals per condition.

Comment 18:

Typos (there are lots). "human families causes in transgenerational shortening"

Response 18:

Thank you for catching this error. We carefully proofread our manuscript and corrected errors throughout.

Comment 19:

Throughout the text many things capitalised (e.g. Transgenerational Epigenetic Inheritance, Paramutation, Telomere Position Effect). Don't capitalise these things.

Response 19:

Thank you for this comment. We have toned down capitalization to be consistent with the literature.

Comment 20:

"oocytes" page 4

Response 20:

Thank you for this comment. We have corrected the error.

Comment 21:

Formatting italics for consistency in Methods "Generation of mNeonGreen::*pot-2* transgenic strain"

Response 21:

Thank you. We have changed our use of italics based on Wormbase recommendations.

Comment 22:

Fig S3E left panel. Line 3,3,3. Should be 1,2,3.

Response 22:

Thank you for pointing out this error. We corrected it.

Comment 23:

In Methods, 5 microlitres expressed as 5uL. Please use greek symbols.

Response 23:

Thank you for catching this. We corrected this laboratory jargon.

Comment 24:

C. elegans nomenclature should be used consistently: 'POT-1 (foci)' instead of 'Pot1 (foci)'

Response 24:

C. elegans has two Pot1 homologs, POT-1 and POT-2, and for experiments where these individual proteins are studied we are careful to use their *C. elegans* names: POT-1 or POT-1::mCherry or POT-2 or mNeonGreen::POT-2. However, the names POT-1 and POT-2 are generally viewed as *C. elegans* jargon by readers in the telomere field, so we refer to foci composed of POT-1 and POT-2 as 'Pot1 foci'. We tried changing this to 'POT1' foci, which would be consistent with mammalian nomenclature and almost consistent with capital letters appropriate for *C. elegans* proteins. However, we felt that 'POT1 foci' would create confusion from *C. elegans* readers, who might think that we simply forgot the dash from *C. elegans* 'POT-1'. Therefore, we decided that 'Pot1 foci' is a better general term that will make this paper easier to understand. And 'Pot1' is proper nomenclature for *S. pombe*, where Pot1 was discovered by Baumann and Cech. We now state: " For simplicity, we hereafter refer to nuclear foci composed of *C. elegans* POT-1::mCherry and mNeonGreen::POT-2 proteins as 'Pot1 foci'. " immediately after we demonstrate that POT-1 and POT-2 foci co-localize.

Reviewer #2 (Remarks to the Author):

Comment 1:

In this study, Lister-Shimauchi et al developed transgenic *C. elegans* lines that express fluorescent proteins that are fused to POT-1 and POT-2, which are telomere binding proteins regulating telomere length, and successfully visualized localization of POT-1 and POT-2 in embryos. By using this system, the authors demonstrated that POT-1::mCherry and mNeonGreen::POT-2 fusion proteins make foci in adult germ cells that vanish in 1-cell embryo, and gradually accumulate again during embryonic development. The authors also found that, in embryos from *pot-2* mutant gametes, POT-1 and POT-2 fusion proteins form numerous foci throughout embryonic development, and very interestingly, this unexpected foci formation phenotype is epigenetically inherited for 6 generations, which is one of the longest types of transgenerational epigenetic inheritance ever documented. In addition, the authors also found that deficiency for POT-1 in gametes results in reduced foci number of POT-1::mCherry in all stages of embryos, and this phenotype is also epigenetically inherited for multiple generations. This study reveals a novel form of transgenerational epigenetic inheritance related to telomeres. I believe that this study provides important new insights into transgenerational epigenetic inheritance.

Overall, experiments are well designed and the impression of this study is very positive. One concern is that transgene expression could alter expression and localization of endogenous genes, and fusion proteins could exhibit unexpected functions. Therefore, the authors should investigate the expression, localization, and function of fusion proteins as well as the effects of transgene expression on endogenous proteins more carefully. Especially, the authors need to show that POT-1::mCherry and mNeonGreen::POT-2 foci indeed are actually at telomeres,

Response 1:

We are sorry but we have not performed DNA FISH to show that the epitope-tagged POT-1 and POT-2 proteins localize to *C. elegans* telomeres. Localization of the POT-1::mCherry transgenic protein to telomeres has previously been demonstrated using end-to-end chromosome fusions that reduced the number of POT-1::mCherry foci by two (Shtessel, 2013). The fact that mNeonGreen::POT-2 foci consistently colocalize with POT-1::mCherry strongly supports the idea that it also localizes to chromosome ends. We tried generating a *pot-1::mKate* fusion protein by knocking mKate into the endogenous *pot-1* locus but were not able to obtain excision of the reporter construct in a manner that would allow for expression of the epitope tagged gene. We do not understand why this occurred, but we have encountered resistance to tagging of endogenous genes for other *C. elegans* genes.

Comment 2:

and that homozygous mNEONnGreen::POT-2 animals do not exhibit the phenotype found in *pot-2* mutant animals.

Response 2:

We asked if telomere length was affected in homozygous *mNeonGreen::pot-2* animals but did not observe a change. This new Southern blotting data shows that *mNeonGreen::pot-2* and *pot-1::mCherry* do not lead to long telomeres, as occurs in both *pot-1* and *pot-2* mutants, indicating that their function is maintained (Supplementary Fig. 2A).

Comment 3:

In addition, if the authors could demonstrate the foci formation of endogenous POT-1 in early embryos from *pot-2* mutant gametes, it would significantly improve the quality of this study.

Response 3:

Unfortunately, there are no available antibodies for either POT-1 or POT-2.

Comment 4:

Minor comments

1. A breeding scheme of each experiment would help readers to understand the manuscript.

Response 4:

Thank you for this suggestion. We have incorporated a general diagram into the paper for clarity (Supplementary Fig. 2C). Most of the experiments in our manuscript are variations on this basic scheme.

Comments 5-7:

2. Some of the results described in the manuscript are missing.

- 1) Page4, line5: high levels of POT-1::mCherry foci in all 1- and 2-cell embryos derived from progeny of *pot-2* animals whose mothers were *pot-2 +/-* heterozygous

2) Page4, line7: heterozygosity for pot-2 does not result in abundant POT-1::mCherry foci in early embryos.

3) Page5, line19: wild-type levels of Pot1 foci were observed at later stages of development.

Response 5-7:

Reviewer #1 had similar comments, and we have adjusted the manuscript to address these concerns.

Reviewer #3 (Remarks to the Author):

Comment 1:

Review of manuscript COMMSBIO-19-0412-T

Key results

The nematode *C. elegans* is an excellent system for studying transgenerational epigenetic inheritance mechanisms, including small-RNA-induced epigenetic silencing and paramutation. In this work, the authors report on a novel case of transgenerational memory in worms, which affects the nuclear foci formed by the POT-1 and POT-2 telomere binding proteins. The work is based on the peculiar observation that single copy POT-1::mCherry has the ability to form developmentally-regulated telomere foci in vivo, which are prominent in the adult germ cells, disappear in 1-cell embryos and are progressively reconstituted during embryonic development. Also POT-2 (tagged with mNeonGreen) has the ability to form foci in vivo, which behave similarly to POT-1 foci. The authors show that mutants deficient for the POT-2 protein lost the ability to regulate the development of such foci, which then persist throughout their life cycle. Interestingly, gametes from pot-2 mutant strains transmit the ability to form POT-1 (and POT-2) foci irrespective of developmental stage. Formation of these foci requires intact POT-1 function. Once exposed to POT-2 deficiency, the memory of deregulated formation of POT1/2 foci acquired from pot-2 mutant gametes persists over many generations, even in the presence of functional POT-2. This transgenerational effect does not appear to be responsive to perturbation in the RNAi machinery or to modulation of chromatin structure.

Validity, originality and significance: Although there is no clear mechanistic explanation for the transgenerational inheritance of the pot-2 -dependent deregulation of POT-1 and POT-2 foci, the work has a robust genetic outline and describes a novel aspect of the complex epigenetic regulation of telomeres. This is of interest not only to specialists in chromosome maintenance and stability, as it uncovers a novel example of acquired epigenetic memory, a phenomenon that persists over many generations, long after the exposure to the initial perturbation. Data & methodology

The experimental outline is sound. Specific comments are provided below.

Appropriate use of statistics and treatment of uncertainties

Statistical analyses should be included for figure 2A and 2B. All other analyses are appropriate.

Response 1:

Thank you for your warm comments. We have added statistics for ANOVA testing.

Comment 2:

Conclusions

Further studies will be required to shed light on the mechanisms governing the described transgenerational epigenetic inheritance of the developmentally regulated formation of POT-1 and POT-2 nuclear foci . As pointed out below, the authors should integrate the discussion of the regulation of POT-1 and POT-2 foci formation with other known aspects of POT-1 and POT-2 biology, such as their ability to modulate telomere recombination and telomere length.

Response 2:

Thank you for this comment. We now state “ It remains possible that high or low levels of Pot1 foci caused by *pot-2* or *pot-1* mutant gametes, respectively, could affect additional functions of Pot1 that include repression of telomere recombination, repression of resection of the C-rich strand of the telomere, or repression of DNA damage checkpoint activation at telomeres ”.

Comment 3:

Suggested improvements

1) Previous work from the same group (Shtessel et al., 2013 DOI: 10.1534/g3.112.004440) has shown that POT-1 forms foci at telomeres. This is particularly clear in pachytene nuclei, in which an average of 12 POT-1::mCherry foci /nucleus are visible, likely identifying the telomeres of six paired homologous chromosomes, while 19 foci in average are visible in mitotic nuclei, compared to the expected 24 foci. To improve clarity, it would be appropriate to include here a brief explanation of what the number of expected foci is, relative to the number of chromosomes. This elucidation is relevant, as there is no actual proof that the POT1::mCherry foci indeed colocalize with telomeres, besides the fact that the number of foci is compatible with the number of telomere ends. No colocalization with a telomere probe is shown.

Response 3:

This is a good point. We now state in the second paragraph of the manuscript “ We previously created a single-copy transgene that expresses POT-1::mCherry and observed nuclear mCherry foci at telomeres of adult *C. elegans* germ cells, as the number of POT-1::mCherry foci in meiotic germ cells corresponded to the expected number of telomeres, which was manipulated using chromosome fusions ”.

Comment 4:

2) The authors previously suggested that telomere clustering may account at least in part for the deviation in comparison to the number of expected foci in wild type cells. Work from the Gasser

group (Ferreira et al., 2013 doi.org/10.1083/jcb.201307181) suggested that telomeres are enriched at the nuclear periphery and tend to be clustered in strains which activated the ALT pathway. Peripheral localization of telomeres appears to be unaffected in *pot-2* mutants, while POT1 is capable of mediating telomere positioning at the nuclear periphery. The authors should discuss the possibility that the number of observed foci may reflect the state of telomere clustering or their positioning within the nucleus. Could POT2-deficiency induce a heritable change in telomere anchoring or mobility that accounts for the increase in POT1 foci observed in later generations? Can they comment as to whether the disappearance of POT1::mCherry and mNeonGreen::POT-2 foci in *pot-1* mutants is related to loss of POT-1-mediated telomere anchoring at the nuclear periphery?

Response 4:

This is a terrific point. We did not address the nuclear position of telomeres in progeny of *pot-1* or *pot-2* mutant gametes experimentally. We did try performing RNAi to *sun-1*, which like *pot-1* helps to anchor telomeres to the nuclear periphery, but did not observe an effect on levels of Pot1 foci. As RNAi knockdown results can be meaningless if negative, we decided not to include this result in our manuscript, even though our positive control RNAi clones worked.

We now show that loss of either *met-2* or *set-25* in gametes is sufficient to deplete Pot1 foci in progeny. As loss of both *met-2* and *set-25* (in a *met-2; set-25* double mutant) does not affect telomere anchoring to the nuclear periphery, we suggest that it is unlikely that POT1 focus levels are related to telomere position. We now state in the Discussion: “ *C. elegans* telomeres have been previously shown to localize to the nuclear periphery in a manner that is lost in *pot-1* mutants but not in *pot-2* mutants or in *met-2; set-25* double mutants ²⁴ .

Localization of heterochromatic transgenes and H3K9me-rich chromosome arms to the nuclear periphery is promoted by either MET-2 or SET-25 alone, but is lost in *met-2; set-25* double mutants ⁴³ . In contrast, loss of either MET-2 or SET-25 alone led to dramatic reductions in Pot1 foci, indicating that the impact of H3K9 methylation at telomeres and other heterochromatic segments of the genome is distinct from telomere localization within the nucleus. It remains possible that deficiency for either POT-1 or POT-2 in gametes causes a heritable change in telomere positioning or mobility that contributes to or is associated with altered levels of Pot1 foci. ”

Comment 5:

3) The quality of the TRF in Supplementary Fig. 3E and F is not optimal for conclusion that telomere shortening actually occurs in the 4 generations explored. The authors should include labels in the TRF lanes to help locate mean telomere length. They should also point out if telomere heterogeneity is observed in *pot-1* and *pot-2* derived strains.

Response 5:

We have modified relevant panels (now Fig. 3C-E) to include a labeled section of the blot for clarity. In our hands, *pot-1* and *pot-2* mutants display similar levels of telomere length

heterogeneity (Shtessel, L. et al. Caenorhabditis elegans POT-1 and POT-2 repress telomere maintenance pathways. G3 3, 305–313 (2013).). However, another study observed differences between *pot-1* and *pot-2* mutant strains (Raices, M. et al. C. elegans telomeres contain G-strand and C-strand overhangs that are bound by distinct proteins. Cell 132, 745–757 (2008).). The Shtessel et al. study used outcrossed *pot-1* and *pot-2* mutant strains, whereas the Raices study used non-outcrossed strains that contain many mutations from chemical mutagenesis. Therefore, it remains unclear if mutation of *pot-1* or *pot-2* cause distinct effects on telomere length. In the current manuscript, we show that when *pot-1* or *pot-2* mutant strains are outcrossed and F2 lines that are wild type for *pot-1* or *pot-2* are isolated, that we observe similar mild reductions in telomere length.

Comment 6:

4) POT-2 deficiency has been shown to affect the amount of telomeric recombination and C-circles, especially at later generations. Is it possible that POT-2 deficiency may exert a dominant effect on telomere recombination, which persists in late generation descendants of *pot-2* gametes?

Response 6:

The elongated telomeres of late-generation *pot-1* and *pot-2* mutants depend on elevated levels of telomerase activity, which is a common property of these mutants that is distinct from their effects on Pot1 focus levels. The Reviewer correctly points out that levels of C-circles become elevated in *pot-2* mutants that have been grown for >20 generations, possibly because of high levels of telomere recombination in early generation *pot-2* mutants leads to long telomeres and high levels of C-circles generation, or possibly because loss of *pot-2* in strains with long telomeres induces telomere trimming via C-circle formation (Fig. 4, Shtessel, 2013). Early-generation *pot-2* mutants with telomeres of moderate length displayed high levels of Pot1 foci, but should have had low levels of C-circles (Shtessel 2013). We did not ask if deficiency for *pot-2* for 4-5 generations would lead to moderate or high levels of C-circles and if descendants of *pot-2* mutant gametes also display this phenotype. We agree that this would be interesting to test in a future study that concerns the consequences of high or low levels of telomeric foci.

Comment 7:

Additional points

-Figure 2A and 2B: statistical analyses for multiple comparisons between classes should be provided

Response 7:

Thank you for this suggestion. We added ANOVA statistics for these comparisons.

Comment 8:

- The labeling of the *pot-2* and *pot-1* derived lines in FIG.S3E and S3F is confusing.

Response 8:

We updated the figure for clarity. Thank you.

Comment 9:

-Genetic crosses in Fig. S1P: check the labeling of the markers in the F1 hermaphrodites

Response 9:

Thank you. We have corrected the labels (now Supplementary Fig. 1E).

Reviewers' Comments:

Reviewer #1:

Remarks to the Author:

The revised manuscript is much improved with additional data and clearer descriptions of the results and conclusions. This is an interesting and intriguing discovery and one of the longest lasting induced transgenerational epigenetic memories. I recommend accepting it for publication with only minor text/figure modifications.

- The authors sometimes use the term 'transgenerational effect' when referring to F2 phenotypes. Strictly this should be an 'intergenerational effect' with transgenerational only used to refer to F3 onwards for inheritance via oocytes.
- It would be useful to have the summary figure (SF3D) as the last main text figure panel.
- Fluorescent foci are not countable in images they show. It would be helpful to show an example higher magnification image as this is one of the main readouts in the manuscript
- spr-5 was reported as having no phenotype in the first submission but now has a phenotype. spr-5 in Figure 3A, shows a substantial reduction of POT-1::mCherry foci. This is in stark contrast to the observation made in the first manuscript (old MS Figure 2K), where maternal spr-5 had no effects on the foci count. Why is this? Has this result been replicated and is robust enough for publication? Also spr-5 is not mentioned in the discussion.
- P.10: 'We found that F1 cross progeny possessed F2 embryos with wild-type levels of POT-1 and POT-2 foci (Supplementary Fig. 2F)'
Only POT-1 foci shown in image
- p.10 'we established a stable trt-1 mutant strain carrying epitope-tagged Pot1 proteins that maintained its telomeres by ALT'
Which epitope? How was this strain made?
- Discussion: 'pot-1 or pot-2 mutant gametes altered the levels of Pot1 foci for multiple generations (diagrammed in Fig. 3F, G).
Shouldn't pot-1 mutants have a phenotype that has returned to normal in the F3 not F7?
- Discussion: 'As loss of H3K9 methyltransferases that have been previously demonstrated to establish silent chromatin domains strongly compromises the creation of Pot1 foci for multiple generations'.
Have the authors tested beyond the F2? The results show F2 embryos (Suppl. 3B) and F1 embryos (3A) derived from the mutant crosses.
- Discussion: "gametes deficient for jmjd-2 were able to induce high Pot1 foci levels for multiple generations." Where is this data shown? Only the effects on F1 are presented in Figure 3A
- One substrate of SET-32 has been recently reported:
Caenorhabditis elegans nuclear RNAi factor SET-32 deposits the transgenerational histone modification, H3K23me3. Schwartz-Orbach L, Zhang C, Sidoli S, Amin R, Kaur D, Zhebrun A, Ni J, Gu SG. Elife. 2020 Aug 17;9:e54309. doi: 10.7554/eLife.54309. PMID: 32804637 Free PMC article.
- Suppl fig 1: 'pot-2 mutants possessing normal telomere lengths, corresponding to F4 embryos in

diagram P'

What/where is diagram P?

- the Southern blots and the text in Figure 3 C and D should be enlarged for legibility.

The authors did not localise the pot-1/pot-2 foci with the telomeres (e.g. via FISH). Instead, they reason with the foci count, as it matches the expected telomere number. However, it still could be that the observed foci also form at non-telomeric regions. This should at least be mentioned.

References. PMIDs 28428426 and 28835928 are striking examples of multi-generation epigenetic memory of reduced heterochromatin repression at H3K9methylated repetitive sequences but are not cited. It strikes me that the underlying mechanisms may be closely related to what is reported here at telomeric loci.

Reviewer #2:

Remarks to the Author:

My comments have been effectively addressed. With the responses to queries and the edits in this version of the manuscript, it is appropriate for publication.

Reviewer #3:

Remarks to the Author:

In the revised version of the manuscript "Gametes deficient for Pot1 telomere binding proteins alter levels of telomeric foci for multiple generations", the authors sufficiently addressed the points raised.

The new version of the manuscript includes additional experiments that expand the repertoire of proteins involved in the deposition of histone marks, which were tested for their involvement in the epigenetic control of Pot1 foci formation. In addition to the previously characterized SPR-5 demethylase, whose loss suppresses Pot1 foci formation, here they provide evidence that deficiencies for the MET-2, SET-25 or the SET-32 methyltransferases are also able to induce a reduction of Pot1 foci, whereas deficiency for the JMJD-2 demethylase induces an increase in Pot1 foci. Although the cascade of events that trigger the disappearance or the persistence of Pot1 foci and the propagation of their status across subsequent generations is unknown, this new data supports the hypothesis that the level of H3K9 methylation at telomeres may be one critical determinant.

I have a few suggestions that would improve the clarity of the manuscript further:

1) Page 7: It would help if the highest clarity was used when indicating the embryos derived from hermaphrodites at F2 or F3 generations.

In the following sentence at Page 7:

Hermaphrodites from this outcrossed pot-2 mutant strain were crossed with males containing the pot1::mCherry transgene, and freshly derived F2 pot-2 -/- mutant hermaphrodites were observed to possess high levels of POT-1::mCherry foci in all 1- and 2-cell F3 embryos (Supplementary Fig. 1F). However, in the diagram in Fig. 1E, most pot-2 -/- individuals are obtained by selecting recombinants FROM F2 hermaphrodites, which are heterozygous for pot-2. In the methods section at page 19 it is clearly stated that pot-2 -/- mutant hermaphrodites are selected among the F3 adult progeny. Please clarify this point in the main text.

2) Page 11: I suggest the following change: "gametes generated by pot-1 and pot-2 mutants" instead

of "pot-1 or pot-2 mutant gametes" to clearly specify that gametes which initiate the transgenerational effect are indeed derived from pot-1 and pot-2 mutant strains. It would help to stress further that the diploid parental genotype that produced the initial gamete (rather than the genotype of the gamete per se) is important for establishing the transgenerational effect (but not for its maintenance over subsequent generations), as explained in the following sentence at page 8: "When males expressing POT-1::mCherry and mNeonGreen::POT-2 were crossed with pot-2(tm1400) mutant hermaphrodites, abundant POT1::mCherry foci were observed in 1- and 2-cell embryos of pot-2 heterozygous F1 cross progeny (diagrammed in Supplementary Fig. 2C). This phenotype persisted for six generations, even for F2 that lacked the pot-2(tm1400) mutation".

Accordingly, in the cross shown in Supplementary Fig. 1E, embryos inheriting pot-2 gametes from F2 heterozygous hermaphrodites, did not exhibit the change in POT-1 foci, as did those inheriting the pot-2 gametes from F3 heterozygous hermaphrodites.

3) Legend of Supplementary Fig. 3: please define the color code used in this figure as well so as to indicate the Pot1 (POT-1 or POT-2) foci 4) Page 14 (discussion): "pot-1 or pot-2 mutant gametes"; see the note above, referred to page 11

Point by point responses for second revision of Lister *et al.* , "Gametes deficient for Pot1 telomere binding proteins alter levels of telomeric foci for multiple generations":

Reviewer #1 (Remarks to the Author):

Comment 1:

The revised manuscript is much improved with additional data and clearer descriptions of the results and conclusions. This is an interesting and intriguing discovery and one of the longest lasting induced transgenerational epigenetic memories. I recommend accepting it for publication with only minor text/figure modifications.

- The authors sometimes use the term 'transgenerational effect' when referring to F2 phenotypes. Strictly this should be an 'intergenerational effect' with transgenerational only used to refer to F3 onwards for inheritance via oocytes.

Response 1:

Thank you for clarifying the distinct nature of the phenotypes we report. We have updated the text to use "intergenerational" with reference to phenotypes that are limited to F1 and F2 progeny of oocytes, and "transgenerational" to refer to phenotypes that persist from the F3 generation onwards. We also include a summary statement that defines the terms intergenerational and transgenerational for the reader in a manner that will allow them to accurately interpret the results and conclusions of our manuscript.

Comment 2:

- It would be useful to have the summary figure (SF3D) as the last main text figure panel.

Response :

Thank you, we have moved the panel to main figure 4.

Comment 3:

- Fluorescent foci are not countable in images they show. It would be helpful to show an example higher magnification image as this is one of the main readouts in the manuscript

Response :

We have added a higher magnification image to the panel.

Comment 4:

- spr-5 was reported as having no phenotype in the first submission but now has a phenotype. spr-5 in Figure 3A, shows a substantial reduction of POT-1::mCherry foci. This is in stark contrast to the observation made in the first manuscript (old MS Figure 2K), where maternal spr-5 had no effects on the foci count. Why is this? Has this result been replicated and is robust enough for publication?

Also *spr-5* is not mentioned in the discussion.

Response :

We discovered that our *spr-5* stock did not have an *spr-5* mutation and ordered a replacement from the *C. elegans* Genetic Stock Center for retesting. We assayed the freshly obtained *spr-5* mutant strain twice, obtaining similar results for both tests.

We have added more discussion on the *spr-5*: “ In addition, the loss of the H3K4 demethylase *spr-5* reduced telomeric Pot1 foci. While SPR-5 may demethylate H3K9⁶⁶, its main activity is demethylation of H3K4, which promotes H3K9 methylation⁴⁶. ”

Comment 5:

- P.10: ‘We found that F1 cross progeny possessed F2 embryos with wild-type levels of POT-1 and POT-2 foci (Supplementary Fig. 2F)’

Only POT-1 foci shown in image

Response :

We have updated this figure panel to include mNeonGreen::*POT-2*.

Comment 6:

- p.10 ‘we established a stable *trt-1* mutant strain carrying epitope-tagged Pot1 proteins that maintained its telomeres by ALT’

Which epitope? How was this strain made?

Response :

We have clarified this section of the text. It now reads “ We utilized progeny of these crosses to establish a stable *trt-1* mutant strain that expressed POT-1::*mCherry* and maintained its telomeres by ALT ”.

Comment 7:

- Discussion: ‘*pot-1* or *pot-2* mutant gametes altered the levels of Pot1 foci for multiple generations (diagrammed in Fig. 3F, G).

Shouldn't *pot-1* mutants have a phenotype that has returned to normal in the F3 not F7?

Response :

Thank you for catching this mistake. We have updated the generation number in the panel (now Fig. 4b).

Comment 8:

- Discussion: ‘As loss of H3K9 methyltransferases that have been previously demonstrated to establish silent chromatin domains strongly compromises the creation of Pot1 foci for multiple generations’. Have the authors tested beyond the F2? The results show F2 embryos (Suppl. 3B) and F1 embryos (3A) derived from the mutant crosses.

Response :

We did not test the methyltransferase progeny further than F2 embryos, so we now note that this data implies that H3K9 methyltransferases affect Pot1 foci for at least two generations but that we have not studied the persistence of this effect. The added text is: "Although we did not observe an effect of small RNA biogenesis factors on Pot1 foci, loss of two H3K9 methyltransferases, MET-2 and SET-25, or loss of the H3K23 methyltransferase SET-32 mimicked the low levels of Pot1 foci observed in response to deficiency for *pot-1* until the F2 generation. We did not test for an effect of these methyltransferase mutations in later generations. " This is a good idea for future experiments.

Comment 9:

- Discussion: "gametes deficient for *jmjd-2* were able to induce high Pot1 foci levels for multiple generations." Where is this data shown? Only the effects on F1 are presented in Figure 3A

Response :

Quantification in F2 embryos is shown in Figure 3a-b. We can see how this would be confusing, as we specified F2 in the figure legend but said only "progeny of animals deficient for *jmjd-2*" in the text. We meant to refer to F2 progeny, and we have now edited the text to clarify this.

Comment 10:

- One substrate of SET-32 has been recently reported:

Caenorhabditis elegans nuclear RNAi factor SET-32 deposits the transgenerational histone modification, H3K23me3. Schwartz-Orbach L, Zhang C, Sidoli S, Amin R, Kaur D, Zhebrun A, Ni J, Gu SG. *Elife*. 2020 Aug 17;9:e54309. doi: 10.7554/eLife.54309. PMID: 32804637 Free PMC article.

Response :

Thank you for pointing out this relevant article. Interestingly, there is evidence of a relationship between H3K23 and H3K9, in which the presence of H3K23me2 is correlated with H3K9me3 and anti-correlated with H3K9me2. In contrast, H3K23me3 correlates with methylated H3K9me2. Although we do not discuss this in our manuscript, this information presents attractive testable hypotheses for future work

Comment 11:

- Suppl fig 1: 'pot-2 mutants possessing normal telomere lengths, corresponding to F4 embryos in diagram P'

What/where is diagram P?

Response :

Thank you for catching this. We have updated the text to state "corresponding to F4 embryos in panel e".

Comment 12:

- the Southern blots and the text in Figure 3 C and D should be enlarged for legibility.

Response :

We have adjusted the size of the Southern blots.

Comment 13:

The authors did not localise the pot-1/pot-2 foci with the telomeres (e.g. via FISH). Instead, they reason with the foci count, as it matches the expected telomere number. However, it still could be that the observed foci also form at non-telomeric regions. This should at least be mentioned.

Response :

This is correct, thank you. The first paragraph of the results section has been modified to clarify this point "While it remains a formal possibility that localization may occur at other places, its specificity to telomeres is supported by a predictable change in foci in response to chromosome fusions".

Comment 14:

References. PMIDs 28428426 and 28835928 are striking examples of multi-generation epigenetic memory of reduced heterochromatin repression at H3K9 methylated repetitive sequences but are not cited. It strikes me that the underlying mechanisms may be closely related to what is reported here at telomeric loci.

Response :

Thank you for this point. We have now included these references in the discussion and pointed out the biochemical parallels where H3K9 methylation can be an integral part of transgenerational epigenetic inheritance. We look forward to testing the possible relationship between transgenerational inheritance reported in either of these papers and telomeric foci. For example, one of these papers discusses impaired DNA replication, and it would certainly be interesting if this were tied to telomeric foci.

Reviewer #2 (Remarks to the Author):

Comment 1:

My comments have been effectively addressed. With the responses to queries and the edits in this version of the manuscript, it is appropriate for publication.

Response :

We appreciate your feedback and assistance with the review process.

Reviewer #3 (Remarks to the Author):

Comment 1:

In the revised version of the manuscript “Gametes deficient for Pot1 telomere binding proteins alter levels of telomeric foci for multiple generations”, the authors sufficiently addressed the points raised.

The new version of the manuscript includes additional experiments that expand the repertoire of proteins involved in the deposition of histone marks, which were tested for their involvement in the epigenetic control of Pot1 foci formation. In addition to the previously characterized SPR-5 demethylase, whose loss suppresses Pot1 foci formation, here they provide evidence that deficiencies for the MET-2, SET-25 or the SET-32 methyltransferases are also able to induce a reduction of Pot1 foci, whereas deficiency for the JMJD-2 demethylase induces an increase in Pot1 foci. Although the cascade of events that trigger the disappearance or the persistence of Pot1 foci and the propagation of their status across subsequent generations is unknown, this new data supports the hypothesis that the level of H3K9 methylation at telomeres may be one critical determinant.

I have a few suggestions that would improve the clarity of the manuscript further:

1) Page 7: It would help if the highest clarity was used when indicating the embryos derived from hermaphrodites at F2 or F3 generations.

In the following sentence at Page 7:

Hermaphrodites from this outcrossed *pot-2* mutant strain were crossed with males containing the *pot1::mCherry* transgene, and freshly derived F2 *pot-2* *-/-* mutant hermaphrodites were observed to possess high levels of POT-1::mCherry foci in all 1- and 2-cell F3 embryos (Supplementary Fig. 1F).

However, in the diagram in Fig. 1E, most *pot-2* *-/-* individuals are obtained by selecting recombinants FROM F2 hermaphrodites, which are heterozygous for *pot-2*. In the methods section at page 19 it is clearly stated that *pot-2* *-/-* mutant hermaphrodites are selected among the F3 adult progeny. Please clarify this point in the main text.

Response :

Thank you for catching this. The main text has been corrected to read “freshly derived F3 *pot-2* *-/-* mutant hermaphrodites”.

Comment 2:

2) Page 11: I suggest the following change: “gametes generated by *pot-1* and *pot-2* mutants” instead of “*pot-1* or *pot-2* mutant gametes” to clearly specify that gametes which initiate the transgenerational effect are indeed derived from *pot-1* and *pot-2* mutant strains. It would help to stress further that the diploid parental genotype that produced the initial gamete (rather than the genotype of the gamete per se) is important for establishing the transgenerational effect (but not for its maintenance over subsequent generations), as explained in the following sentence at page 8: “ When males expressing POT-1::mCherry and mNeonGreen::POT-2 were crossed with *pot-2(tm1400)* mutant hermaphrodites, abundant POT1::mCherry foci were observed in 1- and 2-cell embryos of *pot-2* heterozygous F1 cross progeny (diagrammed in Supplementary Fig. 2C). This phenotype persisted for six generations, even for F2 that lacked the *pot-2(tm1400)* mutation”.

Accordingly, in the cross shown in Supplementary Fig. 1E, embryos inheriting pot-2 gametes from F2 heterozygous hermaphrodites, did not exhibit the change in POT-1 foci, as did those inheriting the pot-2 gametes from F3 heterozygous hermaphrodites.

Response :

This is a terrific point. The previous phrase that gametes deficient for pot-2 is inaccurate because haploid sperm that contain a pot-2 mutation only have an effect when they come from a homozygous mutant animal. We have changed our phrasing with respect to gametes throughout the manuscript.

Comment 3:

3) Legend of Supplementary Fig. 3: please define the color code used in this figure as well so as to indicate the Pot1 (POT-1 or POT-2) foci

Response :

We have added a sentence at the end of the figure legend to explain the colors : “ Red circles indicate POT-1::mCherry, green circles indicate mNeonGreen::POT-2, and yellow circles indicate colocalization of both transgenic proteins. ” This now appears in the legend for figure 4, where the panel now resides.

Comment 4:

4) Page 14 (discussion): “pot-1 or pot-2 mutant gametes”; see the note above, referred to page 11

Response :

We have changed our phrasing with respect to gametes throughout the manuscript, thank you